# Batched Contextual Reinforcement

Bangji Yang [* 1]  Hongbo Ma [* 2]  Jiajun Fan [1]  Ge Liu [1]

## Abstract

Large Language Models (LLMs) employing Chain-of-Thought reasoning achieve strong performance but suffer from excessive token consumption that inflates inference costs. Existing efficiency methods—such as explicit length penalties, difficulty estimators, or multi-stage curricula—either degrade reasoning quality or require complex training pipelines. We introduce **Batched Contextual Reinforcement (BCR)**, a minimalist, single-stage training paradigm that unlocks efficient reasoning through a simple structural modification: training the model to solve $N$ problems simultaneously within a shared context window, rewarded purely by per-instance accuracy. This formulation creates an implicit token budget that yields several key findings: (1) We identify a novel *task-scaling law*: as the number of concurrent problems $N$ increases at inference time, per-problem token usage decreases monotonically—a phenomenon that arises purely at inference, holds for models both before and after training, and is unrelated to accuracy or the training procedure. BCR makes this regime practical by degrading accuracy far more gracefully than baselines as $N$ grows, establishing $N$ as a controllable throughput dimension. (2) BCR challenges the traditional accuracy-efficiency trade-off by demonstrating a "free lunch" phenomenon at standard single-problem ($N = 1$) inference. Across both 1.5B and 4B model families, BCR reduces token usage by 15.8% to 62.6% while consistently maintaining or improving accuracy across five major mathematical benchmarks (e.g., +13.3% on AIME25 for the 4B model). (3) Qualitative analyses reveal emergent self-regulated efficiency, where models autonomously eliminate

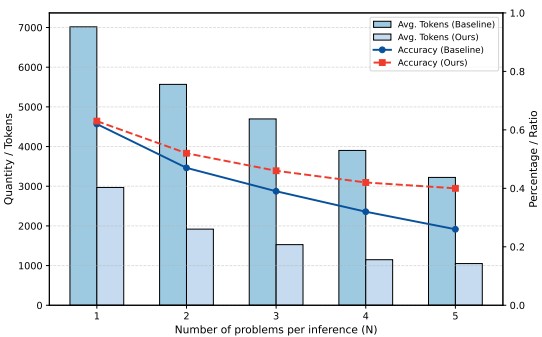

*Figure 1.* **A new scaling dimension: task-level inference scaling on OLYMPIAD.** We vary the number of concurrent problems $N$. **Left axis (bars):** per-problem tokens. **Right axis (lines):** accuracy. The baseline (dark blue) reduces tokens as $N$ grows but suffers accuracy collapse. BCR (light blue) *crystallizes* this efficiency: $\sim$60% token reduction even at $N{=}1$, with graceful accuracy degradation. This reveals a task-scaling law: *more concurrent problems $\Rightarrow$ more efficient reasoning*.

redundant metacognitive loops without explicit length supervision. (4) Crucially, we empirically demonstrate that implicit budget constraints successfully circumvent the adversarial gradients and catastrophic optimization collapse inherent to explicit length penalties, offering a highly stable, constraint-based alternative for length control. These results establish BCR as a highly practical framework, demonstrating how simple structural training incentives can unlock latent high-density reasoning modes in LLMs.

## 1. Introduction

Reinforcement Learning with Verifiable Rewards (RLVR) has become the dominant paradigm for enhancing mathematical reasoning in Large Language Models (Guo et al., 2025; Shao et al., 2024; Yang et al., 2025; OpenAI et al., 2024). By training against objective correctness signals, these methods elicit powerful reasoning behaviors—heuristic search, backtracking, self-correction—that enable small models to tackle competition-level mathematics (Lambert et al., 2025; Li et al., 2025). Yet this capability comes at a steep cost: models optimized on isolated problems develop excessive verbosity, generating redundant reasoning chains that inflate inference latency without proportional accuracy gains (Chen

* Equal Contribution [1]Siebel School of Computing and Data Science, University of Illinois at Urbana-Champaign [2]Department of Computer Science and Technology, Tsinghua University. Correspondence to: Ge Liu <geliu@illinois.edu>.

*Proceedings of the 43rd International Conference on Machine Learning*, Seoul, South Korea. PMLR 306, 2026. Copyright 2026 by the author(s).

et al., 2025a; Shojaee et al., 2025). Existing remedies—explicit length penalties (Aggarwal & Welleck, 2025; Yi et al., 2025; Xiang et al., 2025), auxiliary difficulty estimators (Wu et al., 2025; Xie et al., 2025; Li et al., 2026), and multi-stage curricula (Tan et al., 2026; Song et al., 2025; Liu et al., 2025; Hu et al., 2025)—either degrade reasoning quality, introduce cumbersome pipelines, or require brittle hyperparameter tuning.

This raises a fundamental question: *Can LLMs learn to reason efficiently without any explicit length supervision?*

We answer affirmatively with a strikingly simple observation. When an LLM is asked to solve multiple problems within a single context window, it *spontaneously* compresses its reasoning—using fewer tokens per problem as $N$ increases (Figure 1, dark bars). This reveals a latent capacity for efficient reasoning that standard single-problem training never activates. However, this "passive" compression is fragile: the baseline's accuracy collapses rapidly as $N$ grows (solid blue line), indicating that the model compresses indiscriminately rather than strategically.

Building on this observation, we propose **Batched Contextual Reinforcement (BCR)**, a method that *crystallizes* this latent efficiency into a robust, transferable reasoning policy. The method is minimal: we train the model with GRPO on groups of $N$ problems sharing a fixed token budget, rewarded only by per-instance accuracy. No length penalties, no difficulty estimators, no curriculum scheduling. This creates an implicit information bottleneck: to maximize cumulative accuracy under a shared budget, the model must autonomously discover how to allocate reasoning depth, compress redundant deliberation, and prioritize information density.

BCR yields four principal contributions that collectively advance both the practice and understanding of efficient reasoning:

1. **A task-scaling law for inference efficiency (§4.3).** We discover that the number of concurrent problems $N$ presented within a shared context constitutes a new scaling dimension for inference efficiency: at inference time, increasing $N$ monotonically reduces per-problem token usage, and the reduction grows larger as $N$ increases (Figure 1). This is a purely inference-time phenomenon—it is observed for models both before and after BCR training and is independent of the training group size, reflecting how concurrent problems share a single context rather than any property of the optimization. BCR makes this regime usable in practice: it degrades accuracy far more gracefully than the baseline as $N$ grows, establishing $N$ as a controllable throughput knob—analogous to how batch size scales throughput in classical compute.

2. **A practical single-stage efficient reasoning method (§4.2).** BCR delivers consistent efficiency gains across distinct base model architectures (1.5B and 4B) at standard single-problem ($N=1$) evaluation. It achieves substantial token reductions (from 15.8% up to 62.6%) while consistently maintaining or improving accuracy across five major mathematical benchmarks. Unlike existing approaches that require explicit length penalties (Aggarwal & Welleck, 2025; Yi et al., 2025), auxiliary difficulty estimators (Wu et al., 2025; Li et al., 2026), or cumbersome multi-stage curricula (Tan et al., 2026; Song et al., 2025; Liu et al., 2025), BCR is a single-stage method that modifies only the input structure. This makes it highly accessible, orthogonal to, and composable with existing efficiency techniques.

3. **A "free lunch" that challenges the accuracy-efficiency trade-off (§4.2).** We observe a counterintuitive "free lunch" phenomenon where accuracy actually *increases* across multiple major benchmarks (including AIME25, AMC23, and Minerva) despite significant token compression in both model families. This demonstrates that verbosity in standard reasoning models is not a necessary cost of accuracy but a training artifact: the implicit budget constraint acts as a regularizer that prunes unproductive deliberation loops—repetitive self-checking, redundant strategy exploration, degenerate output sequences—that can actively *harm* reasoning quality.

4. **Emergent self-regulated efficiency (§4.4).** Analysis of reasoning traces reveals that the model progressively develops an intrinsic awareness of concurrent task pressure: it autonomously eliminates metacognitive loops ("*wait, let me re-check...*"), selects optimal strategies directly, and prevents catastrophic degeneration—reducing tokens by up to 92% on individual problems through purely syntactic compression. This emergent behavior provides evidence that LLMs possess latent high-density reasoning modes that resource competition naturally activates, without any explicit efficiency signal.

**Conflict of Interest Disclosure.** The authors declare no financial or other substantive conflicts of interest that could reasonably be perceived to influence this work. None of the authors is employed by an organization that develops any of the models evaluated in this paper.

## 2. Related Work

### 2.1. Reinforcement Learning for Reasoning

RL-based alignment of LLMs has evolved from preference optimization (Schulman et al., 2017; Rafailov et al., 2024) to reasoning enhancement through verifiable rewards. GRPO (Shao et al., 2024) represents a pivotal advance, using

objective verification—such as mathematical ground-truth checking—to compute group-relative advantages without a learned value function. Combined with Long Chain-of-Thought prompting, these methods cultivate sophisticated reasoning behaviors including heuristic search, backtracking, and self-correction (Guo et al., 2025; Lambert et al., 2025; Li et al., 2025). However, optimizing for accuracy on isolated problems incentivizes verbose deliberation: models generate excessive tokens on simple tasks without proportional accuracy gains, a phenomenon termed "overthinking" (Chen et al., 2025a). The resulting low signal-to-noise ratio can actively impair problem-solving performance—extended reasoning chains introduce opportunities for self-contradiction and degenerate outputs (Shojaee et al., 2025; Wang et al., 2025b). BCR addresses this root cause by restructuring the training input—from single-problem to multi-problem batches—creating natural pressure for concise reasoning without modifying the reward function or training algorithm.

### 2.2. Efficient and Adaptive Reasoning

Several lines of work tackle the efficiency bottleneck of Long CoT reasoning, which we organize by their approach to length control.

*Explicit length penalties.* L1 (Aggarwal & Welleck, 2025) and ShorterBetter (Yi et al., 2025) penalize token count directly in the reward function, while ALP (Xiang et al., 2025) scales penalties by problem-solving rate to avoid punishing necessary reasoning. However, as we demonstrate empirically (§4.5.2), explicit penalties create adversarial gradients: the length penalty directly opposes the accuracy reward, leading to an unstable optimization landscape that collapses to degenerate policies producing truncated, incorrect outputs.

*Adaptive reasoning with auxiliary models.* ARM (Wu et al., 2025; Xie et al., 2025) and SelfBudgeter (Li et al., 2026) train auxiliary difficulty estimators that allocate reasoning budgets per problem. Thinker (Chung et al., 2025) introduces mode-switching between fast and slow reasoning paths, while TON (Wang et al., 2025a) and ADR (Zhang et al., 2025) use learned routing mechanisms. While principled, these approaches introduce additional model components, human-designed difficulty taxonomies, and multi-component training procedures that limit scalability and true self-adaptation.

*Multi-stage curricula.* DeepScaleR and FastCuRL (Tan et al., 2026; Song et al., 2025) employ 3–5 stage curricula that progressively extend or compress CoT length. ProRL (Liu et al., 2025) uses an 8-stage pipeline with redesigned length control at each stage, and BroRL (Hu et al., 2025) extends this with aggressive sampling strategies. These approaches achieve strong results but require extensive hyperparameter tuning per stage and are sensitive to curriculum design.

In contrast, BCR requires no auxiliary models, no difficulty labels, no explicit length signal, and no multi-stage scheduling—efficiency emerges purely from inter-problem competition for shared resources. This distinction reflects a deeper insight: rather than *telling* the model to be efficient through reward engineering, BCR creates a training environment where efficiency is *naturally selected* as a consequence of resource competition.

### 2.3. Mathematical Reasoning at Small Scale

Following DeepSeek-R1's (Guo et al., 2025) demonstration that pure RL can elicit complex reasoning, a series of methods have pursued strong mathematical performance at the $\sim$1.5B scale. E3 (Setlur et al., 2025) exploits test-time extrapolation to improve reasoning quality. STILL-3 (Chen et al., 2025b) applies distillation from larger models to compress reasoning capabilities. JustRL (He et al., 2025a) demonstrated that a simple accuracy-only reward suffices for strong reasoning, challenging the necessity of complex training recipes. We build on JustRL's minimalism: BCR modifies only the *input structure*—batching $N$ problems into a single prompt—without changing the reward function, training algorithm, or model architecture. This makes BCR orthogonal to and composable with all of the above methods: one could apply BCR's batched training to any existing pipeline to potentially unlock additional efficiency gains.

## 3. Method

We present **Batched Contextual Reinforcement (BCR)**, a training paradigm that induces efficient reasoning through a single structural modification: training LLMs to solve multiple problems within a shared context window. The key insight is that when $N$ problems compete for a fixed token budget, the model must autonomously discover strategies to compress reasoning—no explicit length penalties or auxiliary difficulty estimators are needed. We describe the formulation (§3.1), optimization (§3.2), reward design (§3.3), and the implicit length control mechanism that distinguishes BCR from all prior approaches (§3.4).

### 3.1. Problem Formulation

Let $\mathcal{D} = \{(q_1, a_1), \ldots, (q_M, a_M)\}$ denote a dataset of mathematical problems where $q_i$ is a problem statement and $a_i$ its ground-truth answer. Standard RLVR approaches optimize a policy $\pi_\theta$ to generate responses $y \sim \pi_\theta(y|q)$ for individual problems. BCR instead formulates optimization over *problem groups*.

**Group Construction.** We partition $\mathcal{D}$ into groups $\mathcal{G} =$

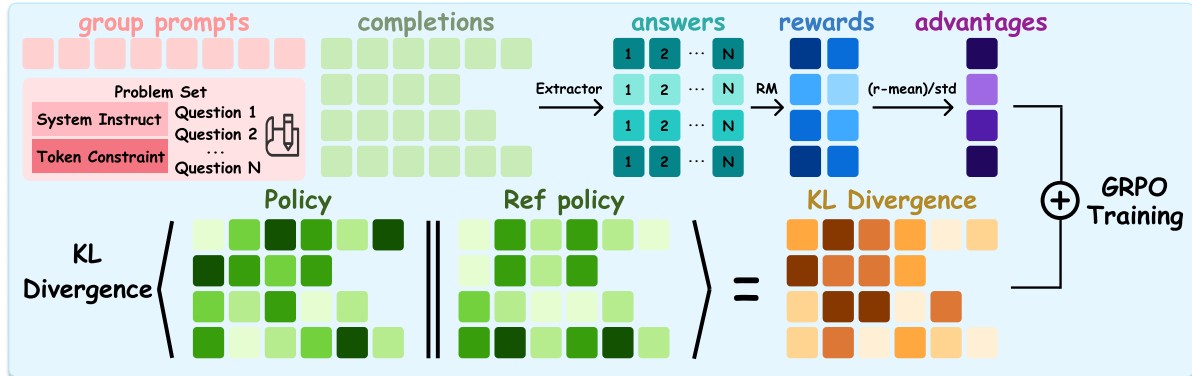

*Figure 2.* **Overview of BCR.** We package $N$ questions into a problem group with a system instruction and shared token budget. The model generates a single completion solving all $N$ problems sequentially. Per-problem answers are extracted via a stack-based parser for accuracy verification, combined with a format reward. Training follows standard GRPO—no length penalties or auxiliary models required.

$\{G_1, \ldots, G_K\}$, where each group

$$G_k = \{(q_{k,1}, a_{k,1}), \ldots, (q_{k,N}, a_{k,N})\}$$

contains $N$ problems. We use stratified sampling based on estimated problem difficulty (proxied by the base model's average reasoning length) to ensure approximately uniform average difficulty per group, while randomizing problem order within each group. This yields $K = M/N$ groups ($K$=3,000 in our main experiments).

**Prompt Construction.** For each group $G_k$, we construct a single prompt $\mathbf{q}_k$ that concatenates all $N$ problems with structural markers:

$$\mathbf{q}_k = [\texttt{SYS}] \oplus \bigoplus_{i=1}^{N} (\texttt{Problem}_i \oplus q_{k,i}) \qquad (1)$$

where $\oplus$ denotes concatenation and $[\texttt{SYS}]$ instructs the model to solve all problems sequentially in a single response (full template in Appendix A). The structured format enables precise per-problem answer extraction while maintaining global coherence across the completion.

### 3.2. Training with GRPO

We optimize the policy using Group Relative Policy Optimization (GRPO) (Shao et al., 2024). Given a reference policy $\pi_{\text{ref}}$ (the base model), we maximize:

$$\max_{\theta} \mathbb{E}_{G \sim \mathcal{G}} \mathbb{E}_{y \sim \pi_\theta(\cdot|\mathbf{q}_G)} [R(y, G) - \beta \mathbb{D}_{\text{KL}}(\pi_\theta \| \pi_{\text{ref}})] \quad (2)$$

where $R(y, G)$ is the group reward and $\beta$ controls KL regularization. For each group $G$, we sample $S$ candidate completions $\{y^{(1)}, \ldots, y^{(S)}\}$ and compute group-relative advantages:

$$A(y^{(j)}, G) = R(y^{(j)}, G) - \frac{1}{S} \sum_{s=1}^{S} R(y^{(s)}, G) \qquad (3)$$

This formulation encourages the model to discover strategies that outperform the average response quality within each group, creating natural selection pressure for efficient, high-reward completions.

### 3.3. Reward Design

The reward decomposes into accuracy and format components:

$$R(y, G) = w_{\text{acc}} \cdot r_{\text{acc}}(y, G) + w_{\text{fmt}} \cdot r_{\text{fmt}}(y) \qquad (4)$$

**Accuracy Reward.** We extract $N$ candidate answers $\{\hat{a}_1, \ldots, \hat{a}_N\}$ from completion $y$ using a stack-based parser (Appendix A.2) and verify each against the ground truth:

$$r_{\text{acc}}(y, G) = \frac{1}{N} \sum_{i=1}^{N} \mathbb{I}[\text{verify}(\hat{a}_i, a_i)] \qquad (5)$$

where $\text{verify}(\cdot, \cdot)$ employs symbolic equivalence checking with fallback to string and numeric comparison.

**Format Reward.** We verify that each answer appears in the designated structured format:

$$r_{\text{fmt}}(y) = \mathbb{I} \left[ \bigwedge_{i=1}^{N} \text{hasFormat}(y, i) \right] \qquad (6)$$

where $\text{hasFormat}(y, i)$ checks that the completion contains "Answer$i$: \boxed{...}" in the appropriate section.

Crucially, **no length-related reward component is used**. Efficiency emerges entirely from the implicit constraint described next.

### 3.4. Implicit Length Control via Token Budget

The central mechanism of BCR is an *implicit information bottleneck*. We impose a fixed token budget $B_{\text{max}}$ for the

entire completion (e.g., $B_{\max} = 5{,}120$ for $N{=}3$ problems). Within this budget, the model must solve all $N$ problems—verbose reasoning on early problems directly reduces the budget available for later ones, potentially truncating their solutions and yielding zero accuracy reward.

This creates a fundamentally different optimization landscape than explicit length penalties:

- **No per-token punishment.** Within the budget, every token is "free"—the model is never penalized for generating tokens, only for failing to solve problems. This avoids the adversarial gradients that cause training collapse with explicit penalties (§4.5.2).
- **Inter-problem competition.** The $N$ problems compete for a shared resource, creating implicit pressure to allocate tokens strategically: use fewer tokens on easier problems to preserve budget for harder ones.
- **Emergent adaptive reasoning.** The model learns to calibrate reasoning depth to problem difficulty *without any difficulty signal*—the budget constraint alone induces this self-regulation.

We show in §4 that this implicit mechanism substantially outperforms explicit length penalties, and that the efficiency it induces transfers fully to single-problem ($N{=}1$) evaluation—demonstrating that BCR fundamentally reshapes the model's reasoning policy rather than merely adapting to batched inference.

# 4. Experiments

We organize experiments to validate each of our five contributions. After describing the setup (§4.1), we present main results demonstrating BCR's efficiency and the "free lunch" phenomenon at standard $N{=}1$ inference (§4.2), the task-scaling law across varying $N$ (§4.3), emergent self-regulated efficiency through qualitative analysis (§4.4), and ablation studies comparing implicit vs. explicit length control (§4.5).

## 4.1. Experimental Setup

**Training.** We implement BCR using GRPO via the TRL library (von Werra et al., 2020), extended with group prompt construction and stack-based answer extraction. Training data consists of 3,000 balanced groups from DeepMath-103K (He et al., 2025b), each containing $N{=}3$ problems with stratified difficulty sampling. We instantiate BCR on two starting points: **JustRL-DeepSeek-1.5B** (He et al., 2025a) and **Qwen3-4B-Thinking-2507** (Yang et al., 2025); this lets us test whether BCR transfer holds from a 1.5B RL baseline to a stronger 4B reasoning model. Full hyperparameters are in Appendix A.1.

**Evaluation.** We evaluate on five benchmarks spanning diverse difficulty levels: AIME 2025 (Balunović et al., 2025),

AMC 2023 (Li et al., 2024), Minerva Math (Lewkowycz et al., 2022), MATH-500 (Hendrycks et al., 2021), and Olympiad (He et al., 2024). All evaluations use temperature 0.6, top-$p$ sampling ($p{=}0.9$), and a maximum generation length of 32,768 tokens. We report accuracy (%) and average generated tokens per problem (excluding prompts).

## 4.2. Main Results: Efficiency at Standard Inference

Table 1 compares BCR against baselines at standard $N{=}1$ inference—the setting where each problem is solved independently, exactly as in conventional evaluation. This is the strictest test of whether BCR's efficiency generalizes beyond the batched training regime.

**Efficiency transfers to single-problem inference.** Although BCR is trained with $N{=}3$, the learned efficiency generalizes fully to $N{=}1$ in both model families: compared with JustRL-deepseek-1.5B, token usage drops by 39.8–62.6% across all five benchmarks; compared with Qwen3-4B-Thinking-2507, token usage drops by 15.8–31.8% across the same benchmarks. This is not a test-time adaptation that requires batched inputs—the model has internalized a fundamentally more efficient reasoning policy that persists regardless of inference format.

**The "free lunch" phenomenon.** For the JustRL-1.5B pair, the free lunch appears on two benchmarks: AMC23 (+2.5) and Minerva (+5.1), both with large token reductions. For the Qwen3-4B pair, it appears on all five benchmarks: +13.3 (AIME25), +2.5 (AMC23), +2.2 (Minerva), +0.4 (MATH-500), and +1.9 (Olympiad), while token usage decreases by 15.8–31.8% across all five. This counterintuitive result suggests that the implicit budget constraint acts as a regularizer: by pruning unproductive deliberation loops—repetitive self-checking, redundant strategy re-exploration, degenerate output sequences—the model actually reasons more reliably. Verbosity, it appears, is not merely wasteful but can be actively harmful to reasoning quality.

**Comparison with existing approaches.** Length-controlled models (BroRL, e3) achieve efficiency through explicit penalties or multi-stage curricula with 3–8 stages. BCR matches or exceeds their token efficiency with a single-stage approach and no length supervision whatsoever. Adaptive reasoning models (ARM, Thinker) achieve extreme brevity but suffer catastrophic accuracy drops (ARM: 3.3% on AIME25; Thinker: 0.0%), confirming that naive length minimization destroys reasoning capability. BCR avoids this failure mode because the implicit budget does not penalize token generation—it only creates competition among problems for shared resources.

To further visualize this efficiency-accuracy trade-off, Figure 3 illustrates the Pareto frontier trajectories during training on the Minerva benchmark. Both BCR-JustRL and

*Table 1.* **Main results across five benchmarks at $N{=}1$ inference.** We report accuracy (%) and average tokens per problem. **Bold** indicates best in column. BCR improves efficiency in both model families: vs. JustRL-deepseek-1.5B, token usage drops by **39.8–62.6%** with accuracy gains on 3/5 benchmarks; vs. Qwen3-4B-Thinking-2507, token usage drops by **15.8–31.8%** with accuracy gains on 5/5 benchmarks. This demonstrates that efficiency learned through multi-problem training transfers fully to standard single-problem evaluation.

| Model | AIME25 | | AMC23 | | Minerva | | MATH-500 | | Olympiad | |
|---|---|---|---|---|---|---|---|---|---|---|
| | Acc↑ | Tok↓ | Acc↑ | Tok↓ | Acc↑ | Tok↓ | Acc↑ | Tok↓ | Acc↑ | Tok↓ |
| **General LLM** | | | | | | | | | | |
| Qwen3-1.7B | 40.0 | 15,796 | 85.0 | 8,157 | 57.7 | 5,714 | 90.2 | 4,621 | 67.3 | 9,395 |
| **Math LLM** | | | | | | | | | | |
| STILL-3-1.5B | 33.3 | 13,512 | 72.5 | 5,706 | 46.7 | 5,678 | 84.6 | 4,280 | 55.2 | 9,755 |
| **Math LLM w/ Length Control** | | | | | | | | | | |
| BroRL-1.5B | 36.9 | 5,435 | 81.0 | 4,120 | 49.1 | 4,205 | 92.1 | 2,996 | 61.5 | 4,352 |
| e3-1.7B | 46.7 | 11,804 | 85.0 | 7,154 | 54.4 | 5,201 | 91.0 | 3,774 | 67.1 | 7,515 |
| **Math LLM w/ Adaptive Reasoning** | | | | | | | | | | |
| ARM-3B | 3.3 | 2,050 | 42.5 | **703** | 27.9 | 3,849 | 58.4 | 1,222 | 31.9 | 4,534 |
| Thinker-Q1.5B | 0.0 | 812 | 32.5 | 828 | 23.5 | 777 | 63.4 | **760** | 25.2 | **813** |
| **Baselines and Ours: JustRL-deepseek-1.5B** | | | | | | | | | | |
| DeepSeek-R1-Distill-Qwen-1.5B | 30.0 | 15,863 | 67.5 | 9,702 | 48.5 | 7,575 | 85.4 | 5,409 | 53.4 | 11,599 |
| JustRL-deepseek-1.5B | 40.0 | 8,482 | 85.0 | 5,713 | 43.4 | 5,413 | 91.4 | 3,099 | 62.1 | 7,017 |
| **BCR-JustRL-1.5B (Ours)** | 33.3 | **3,173** | 87.5 | **2,637** | 48.5 | **2,494** | 87.6 | **1,868** | 62.9 | **2,969** |
| $\Delta$ *vs.* JustRL | −6.7 | −62.6% | +2.5 | −53.8% | +5.1 | −53.9% | −3.8 | −39.8% | +0.8 | −58.0% |
| **Baselines and Ours: Qwen3-4B-Thinking-2507** | | | | | | | | | | |
| Qwen3-4B-Thinking-2507 | 70.0 | 20,773 | 97.5 | 10,457 | 66.2 | 4,576 | 97.0 | 5,136 | 83.3 | 13,069 |
| **BCR-Qwen3-4B (Ours)** | **83.3** | 17,498 | **100.0** | 7,128 | **68.4** | 3,338 | **97.4** | 3,713 | **85.2** | 10,717 |
| $\Delta$ *vs.* Qwen3-4B | +13.3 | −15.8% | +2.5 | −31.8% | +2.2 | −27.1% | +0.4 | −27.7% | +1.9 | −18.0% |

BCR-Qwen models consistently push the frontier towards lower token consumption without sacrificing accuracy. Trajectories for other benchmarks are provided in the Appendix.

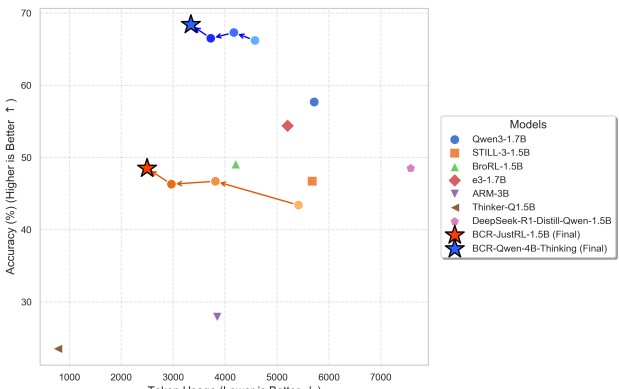

*Figure 3.* **Efficiency-Accuracy Pareto Frontier on Minerva.** The trajectories show checkpoint evaluations during the training process. The final models (stars) demonstrate that BCR consistently pushes the Pareto frontier significantly toward higher accuracy and lower token usage for both model families.

### 4.3. Task-Scaling Law: More Tasks, More Efficiency

A central contribution of this work is the discovery that the number of concurrent inference tasks $N$ constitutes a new scaling dimension for reasoning efficiency. We evaluate both models under $N \in \{1, 2, 3, 4, 5\}$, where $N$ problems are solved simultaneously in a single context window.

Table 2 reveals a clear *task-scaling law*: as $N$ increases, BCR-trained models become progressively more token-efficient while maintaining substantially higher accuracy than the baseline. Three specific patterns emerge.

**Graceful degradation under pressure.** The baseline's accuracy collapses precipitously under concurrent load: on AMC23, it drops from 85.0% ($N{=}1$) to 22.5% ($N{=}5$)—a 74% relative decline. BCR degrades far more gracefully: 87.5% → 50.0%, a 43% relative decline. This robustness directly reflects BCR's training objective: the model has learned to *allocate cognitive resources adaptively* across concurrent problems, compressing strategically rather than indiscriminately. The baseline, never having encountered resource competition during training, lacks this adaptive capacity entirely.

**Widening efficiency advantage.** The relative efficiency

*Table 2.* **Task-scaling law: accuracy (%) and per-problem tokens under** $N\times$ **concurrent inference.** BCR maintains superior efficiency across all group sizes and degrades far more gracefully than the baseline. The efficiency gap *widens* as $N$ increases, establishing a favorable scaling relationship. See Appendix B for MATH-500 and Olympiad.

| $N$ | Model | AIME25 | | AMC23 | | Minerva | |
|---|---|---|---|---|---|---|---|
| | | Acc↑ | Tok↓ | Acc↑ | Tok↓ | Acc↑ | Tok↓ |
| 1× | JustRL-1.5B | **40.0** | 8,482 | 85.0 | 5,713 | 43.4 | 5,413 |
| | BCR (Ours) | 33.3 | **3,173** | **87.5** | **2,637** | **48.5** | **2,494** |
| 2× | JustRL-1.5B | 23.3 | 7,495 | 75.0 | 4,697 | 33.8 | 4,141 |
| | BCR (Ours) | **30.0** | **2,371** | **83.3** | **1,705** | **37.5** | **1,679** |
| 3× | JustRL-1.5B | 20.0 | 4,095 | 55.0 | 5,429 | 24.6 | 3,885 |
| | BCR (Ours) | 20.0 | **1,279** | **72.5** | **1,376** | **31.6** | **1,287** |
| 4× | JustRL-1.5B | 20.0 | 4,618 | 47.5 | 3,000 | 23.5 | 2,880 |
| | BCR (Ours) | **26.7** | **1,142** | **57.5** | **1,186** | **32.7** | **956** |
| 5× | JustRL-1.5B | **26.7** | 2,458 | 22.5 | 4,584 | 19.1 | 3,126 |
| | BCR (Ours) | 16.7 | **930** | **50.0** | **1,013** | **30.9** | **826** |

gap between BCR and the baseline *grows* with $N$. At $N=1$, BCR uses 63% fewer tokens on AIME25; at $N=4$, the gap widens to 75%. This amplification effect reveals that BCR's learned compression strategies become increasingly valuable under tighter resource constraints—precisely the regime relevant for cost-sensitive deployment where throughput matters most.

$N$ **as a controllable inference knob.** These results establish $N$ as a new inference-time parameter for trading throughput against accuracy. A practitioner can increase $N$ to process more problems per API call with predictable, graceful accuracy trade-offs—analogous to how batch size scales throughput in classical compute. This task-scaling law is, to our knowledge, the first systematic characterization of how concurrent problem count affects reasoning efficiency, and it opens a new dimension for optimizing LLM deployment costs.

### 4.4. Emergent Self-Regulated Efficiency

The quantitative results above establish *what* BCR achieves; we now examine *how*. Analysis of individual reasoning traces reveals that BCR induces systematic compression mechanisms that operate at the syntactic level—eliminating verbose patterns without removing any mathematical reasoning steps. This section provides qualitative evidence that the model develops an intrinsic awareness of resource pressure and autonomously adapts its reasoning style.

#### 4.4.1. QUALITATIVE ANALYSIS

In this subsection (4.4.1), the analysis is based on the **JustRL-DeepSeek-1.5B** vs. **BCR-JustRL-1.5B** pair to isolate behavior changes induced by BCR under the same 1.5B backbone. More results, including **BCR-Qwen3-4B-**

**Thinking-2507** quantitative breakdowns and extended comparisons, are provided in Appendix C.3.

Table 3 presents two representative examples that illustrate the systematic compression mechanisms BCR induces. Our full qualitative analysis (Appendix C) identifies four distinct mechanisms across seven examples:

**Metacognitive loop elimination.** The baseline frequently interrupts correct reasoning with self-verification spirals—"*Wait wait wait…*" consumes thousands of tokens without contributing new mathematical content. These loops represent the model second-guessing calculations that were already correct. BCR-trained models proceed linearly: once a valid approach is identified, reasoning flows to the conclusion without unnecessary self-doubt. Crucially, BCR models still verify when *mathematically necessary* (e.g., checking boundary conditions); they simply avoid *redundant* re-verification of already-correct steps.

**Direct strategy selection.** Baseline models often explore multiple solution strategies before committing, even when the first approach is correct. BCR models apply the most effective strategy immediately, suggesting that multi-problem training sharpens problem recognition: the model learns to identify optimal solution paths faster because wasting tokens on suboptimal strategies for one problem reduces the budget available for subsequent problems.

**Prevention of catastrophic degeneration.** On hard problems (Example 2), the baseline exhausts its 32K token budget and degenerates into repetitive, non-mathematical character sequences—a known failure mode of extended autoregressive generation (Chen et al., 2025a). BCR solves the same problem in 2,692 tokens (91.8% reduction) with a direct, insightful solution. The implicit budget constraint

*Table 3.* **Qualitative comparison.** BCR eliminates verbose metacognitive loops and missing structured logic while preserving all essential reasoning steps. **Green** highlights efficient reasoning; *red* marks verbose or failed patterns. See Appendix C for seven additional examples with detailed analysis.

---

**Example 1: Sudoku-Style Counting (Qwen AIME 2025)** (43.2% token reduction)
**Problem:** Count the valid $3 \times 9$ Sudoku-style fillings and write the total as $p^a q^b r^c s^d$; compute $pa + qb + rc + sd$.
**Ground Truth:** 81

| **Qwen3-4B-Thinking-2507 (31,362 tokens)** | **BCR-Qwen3-4B (17,805 tokens)** |
|---|---|
| *Uses $N = 9! \cdot 2^3 \cdot (3!)^3$ and factors this incomplete count. Most extra tokens are spent re-describing the grid geometry and block layout before the counting argument actually starts.* | **Adds the missing structural multiplier: 56 feasible middle-block assignments.** $N = 9! \cdot 56 \cdot (3!)^3 \cdot (3!)^3$. **Thus $N = 2^{16} \cdot 3^{10} \cdot 5 \cdot 7^2$, so $2 \cdot 16 + 3 \cdot 10 + 5 + 14 = 81$.** |
| *The actual error is structural: the trace never explicitly counts the 56 feasible middle-block assignments, so a long derivation is built on the wrong combinatorial factor. Concludes $N = 2^{13} \cdot 3^7 \cdot 5 \cdot 7$, hence $2 \cdot 13 + 3 \cdot 7 + 5 + 7 = 59$.* | |
| **Answer:** $\boxed{59}$ ✗ | **Answer:** $\boxed{81}$ ✓ |

---

**Example 2: Number Theory — Roots of Unity (BRUMO 2025)**
**Problem:** What is the smallest positive integer $n$ such that $z^n - 1$ and $(z - \sqrt{3})^n - 1$ share a common complex root?
**Ground Truth:** 12    **Baseline:** 32,677 tokens    **Ours:** 2,692 tokens ($-91.8\%$)

| **JustRL-DeepSeek-1.5B (Baseline)** | **BCR-JustRL-1.5B (Ours)** |
|---|---|
| *[Model generates over 32,000 tokens]* 
 *[Output degenerates into repetitive patterns:]* 
 `...  2 2 1 11 1 1 1 0, , 11 1 , (1 1` 
 `...` 
 *[No valid answer extracted]* | Let $\alpha$ be a common root. Then $\alpha^n = 1$ and $(\alpha - \sqrt{3})^n = 1$. 
 **If $\alpha = e^{i\pi/6}$, then $\alpha - \sqrt{3} = e^{i \cdot 5\pi/6}$.** 
 Order of $e^{i\pi/6}$ is 12. Order of $e^{i \cdot 5\pi/6}$ is also 12. 
 Verify: $\alpha^{12} = e^{i \cdot 2\pi} = 1$ ✓ 
 **Smallest such $n$ is 12.** |
| **Answer:** None ✗ | **Answer:** $\boxed{12}$ ✓ |

---

trains the model to commit to promising approaches rather than engaging in unbounded exploration that leads to degenerate outputs.

These patterns reveal that BCR induces an *intrinsic awareness of resource pressure*: the model autonomously learns that tokens are scarce and must be allocated to high-value reasoning steps. No prompt mentions efficiency; no reward penalizes verbosity. This self-regulation emerges purely from the structural incentive of multi-problem competition—evidence that LLMs possess latent high-density reasoning modes that resource competition naturally activates.

### 4.5. Ablation Studies

We conduct ablations along two axes: training group size (§4.5.1), the comparison between implicit and explicit length control (§4.5.2). More axes are provided in the Appendix.

#### 4.5.1. TRAINING GROUP SIZE ($N$ SWEEP)

Table 4 evaluates models trained with $N \in \{3, 4, 5\}$, all evaluated at $N=1$. Performance is remarkably stable across training group sizes, with accuracy varying by $<6\%$ and token usage by $<20\%$. This robustness confirms that the implicit competition mechanism operates effectively across group sizes and that practitioners can select $N$ based on infrastructure constraints without risking significant performance degradation.

**Optimal group size depends on benchmark difficulty.** On the easier AMC23 benchmark, both $N=3$ and $N=5$ achieve 87.5% accuracy with similar token efficiency. On the harder AIME25 benchmark, $N=4$ achieves the best accuracy (33.3%) and lowest token count (4,447), suggesting that moderate compression pressure benefits harder tasks by encouraging more focused reasoning. On Minerva, $N=3$ achieves both the highest accuracy (46.3%) and lowest token usage (2,967). The complete five-benchmark analysis (Appendix B) confirms these patterns: $N=3$ provides the most consistently strong accuracy-efficiency balance, which we adopt as our default. Importantly, the stability across $N$ values means that suboptimal choices of $N$ incur only minor performance costs, making BCR practical to deploy without extensive hyperparameter search.

*Table 4.* **Training group size ablation.** All models trained for 300 steps and evaluated at $N{=}1$. $N{=}3$ provides the best accuracy-efficiency trade-off. See Appendix B for MATH-500 and Olympiad.

| $N$ | AIME25 | | AMC23 | | Minerva | |
|---|---|---|---|---|---|---|
| | acc. | tok. | acc. | tok. | acc. | tok. |
| 3 | 30.0 | 5408 | **87.5** | 2722 | **46.3** | **2967** |
| 4 | **33.3** | **4447** | 77.5 | 3466 | 44.5 | 3107 |
| 5 | 30.0 | 5852 | **87.5** | **2671** | 41.9 | 3040 |

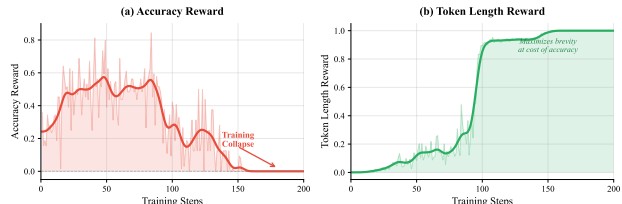

*Figure 4.* **Implicit vs. Explicit length control.** The explicit length penalty settings (both 2-1-1 and 5-1-1) exhibit catastrophic training collapse with rapidly negative accuracy rewards while maximizing length rewards. Our Acc+Format setting (implicit budget) achieves stable optimization, confirming that **hard constraints outperform soft penalties**.

### 4.5.2. IMPLICIT VS. EXPLICIT LENGTH CONTROL

A critical question is whether the implicit budget mechanism is truly necessary, or whether a simpler explicit length penalty would suffice. We compare our implicit budget approach against explicit length penalty configurations:

- **Acc+Format** (ours): $w_{acc}{=}2.0$, $w_{fmt}{=}1.0$; implicit length control via fixed token budget.
- **Explicit Penalty (2-1-1 & 5-1-1)**: We append an explicit penalty $r_{len} = -|y|/\text{max\_len}$ with weight $w_{len}{=}1.0$, evaluating both standard ($w_{acc}{=}2.0$) and high ($w_{acc}{=}5.0$) accuracy weights to ensure robust comparison.

Figure 4 reveals a stark divergence. Both explicit penalty settings (2-1-1 and 5-1-1) induce *catastrophic training collapse*: the model learns to minimize tokens aggressively, maximizing the length reward but driving accuracy to zero and producing truncated, degenerate outputs. This occurs because explicit penalties create *adversarial gradients*—every generated token incurs punishment, even tokens essential for correct reasoning. The gradient signal from the length penalty directly opposes the gradient from the accuracy reward, creating an unstable optimization landscape that collapses to a degenerate policy.

Our implicit approach avoids this entirely: rewards improve monotonically throughout training because the model is never punished for generating tokens within the budget. Efficiency emerges as a *byproduct* of accuracy optimization under resource constraints, not as a competing objective. The key distinction is fundamental: a fixed budget is a *constraint* on the optimization problem, not an additional *penalty term* in the reward function.

## 5. Conclusion

We present **Batched Contextual Reinforcement (BCR)**, a minimalist training paradigm that induces efficient reasoning through a purely structural modification: training LLMs to solve multiple problems within a shared token budget. BCR requires no length penalties, no difficulty esti-

mators, and no multi-stage curricula—efficiency emerges from implicit resource competition alone.

Our work makes four contributions. First, we discover a *task-scaling law*: presenting more problems $N$ concurrently at inference time monotonically reduces per-problem token usage, with the reduction growing as $N$ increases. This is a purely inference-time phenomenon that holds for models both before and after training and is independent of accuracy and the training procedure; BCR makes it deployable by degrading accuracy far more gracefully than baselines, establishing $N$ as a controllable throughput knob. Second, BCR provides a practical single-stage method with consistent $N{=}1$ efficiency gains across two base models: 39.8–62.6% token reduction vs. JustRL-deepseek-1.5B and 15.8–31.8% token reduction vs. Qwen3-4B-Thinking-2507, without any length supervision, making it orthogonal to and composable with existing efficiency techniques. Third, we observe a "free lunch" in both model families: for JustRL-1.5B, accuracy improves on two benchmarks (AMC23 and Minerva); for Qwen3-4B, accuracy improves on all five benchmarks, while token usage is reduced throughout, challenging the assumption that efficiency and accuracy are inherently at odds: verbosity in standard reasoning models is a training artifact, not a cost of accuracy. Fourth, we observe *emergent self-regulated efficiency*: without any explicit efficiency signal, the model eliminates metacognitive loops, selects strategies directly, and prevents catastrophic degeneration—evidence that LLMs possess latent high-density reasoning modes activated by structural incentives.

**Limitations and future work.** BCR has been validated on mathematical reasoning at the 1.5B and 4B scales. Extending to larger models (7B–70B), other reasoning domains (code, scientific, and multi-modal reasoning), and other RL algorithms (PPO, DAPO) are natural next steps, as is a theoretical account of the task-scaling law and its behavior under heterogeneous task mixtures. More broadly, our findings suggest that capabilities elicited through complex training procedures may instead be latent properties accessible via simple structural modifications to the training environment.

## Acknowledgements

This research is partially supported by research funding from Amazon-Illinois Center on AI for Interactive Conversational Experiences (AICE).

## Impact Statement

This work aims to make the reasoning of large language models more computationally efficient. By training models to solve multiple problems under a shared token budget, BCR substantially reduces the number of tokens generated per problem while maintaining or improving accuracy. The most direct positive impact is environmental and economic: lower token consumption translates into reduced energy use, lower inference latency, and lower deployment cost, which can broaden access to capable reasoning models for resource-constrained users and institutions. Because BCR is a single-stage method that modifies only the input structure, it is inexpensive to adopt and composable with existing efficiency techniques, further lowering the barrier to sustainable deployment.

Our experiments are confined to mathematical reasoning benchmarks, where answers are objectively verifiable. We caution that aggressive compression of reasoning traces could, in other settings, reduce the interpretability of intermediate steps or omit reasoning that a human reviewer might wish to audit; practitioners deploying compressed-reasoning models in high-stakes domains should validate that brevity does not come at the cost of transparency or reliability. BCR shares the general risks of any method that improves LLM capability or reduces its cost, including the potential for misuse, but it does not introduce risks beyond those already present in the underlying models. We do not foresee any specific additional societal harms that must be highlighted here.

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

# A. Implementation Details

## A.1. Training Configuration

**Base Model.** We initialize from JustRL-DeepSeek-1.5B, a 1.5B parameter model pretrained with standard GRPO (Shao et al., 2024) on mathematical reasoning tasks. This baseline derives from DeepSeek-R1-Distill-Qwen-1.5B and exhibits strong reasoning capabilities but lacks efficiency optimization—making it an ideal testbed for studying whether BCR can induce efficiency without sacrificing existing reasoning ability. Importantly, JustRL was itself trained with a pure accuracy reward (no length penalties), which means any efficiency gains from BCR cannot be attributed to residual length supervision in the base model. We additionally train BCR from **Qwen3-4B-Thinking-2507**. The reward formulation, grouped-prompt construction, answer extraction, and evaluation pipeline remain the same as in the JustRL-based setting for direct comparability. In our logged Qwen3 runs, we keep the same learning-rate scale ($5 \times 10^{-6}$) and use `max_completion_length`=8000 to better accommodate longer intermediate traces from the 4B model under grouped reasoning.

**System Prompt.** During grouped training, we prepend the following system instruction:

```
[system]
You are an expert mathematics tutor.
Your task is to solve **multiple** math problems sequentially in a single response.
Please strictly follow these rules:
1. Use Markdown headers (### Problem X) to separate each problem.
2. For each problem, show detailed step-by-step reasoning, then immediately put
   the final answer right after the reasoning.
3. Put the final answer for each problem immediately after solving it, in the
   following format:
   After Problem 1: Answer1: \boxed{...}
   After Problem 2: Answer2: \boxed{...}
   After Problem 3: Answer3: \boxed{...}
4. Each answer should appear right after its corresponding problem's reasoning,
   before moving to the next problem.
5. Do not include any other text in your response.
```

**Prompt Template.** The model produces answers in a structured format that enables precise per-problem extraction while maintaining global coherence:

```
### Problem 1
[detailed reasoning steps]
Answer1: \boxed{...}

### Problem 2
[detailed reasoning steps]
Answer2: \boxed{...}

### Problem N
[detailed reasoning steps]
AnswerN: \boxed{...}
```

This template serves two purposes: (1) the explicit section markers (`### Problem` $i$) enable our stack-based parser to reliably segment the completion into per-problem reasoning traces, and (2) the `Answer`$i$`: \boxed{...}` format ensures answers are unambiguously demarcated for verification. During training, the format reward $r_{\text{fmt}}$ enforces adherence to this structure. We chose the markdown-style headers (`###`) because preliminary experiments showed higher format compliance compared to alternatives such as numbered lists or XML-style tags. The model quickly learns to produce well-structured outputs, with format compliance reaching >98% within the first 50 training steps.

**Hyperparameters.** Our main experiments use the following configuration:

- **Training duration:** 3 epochs over 3,000 groups

- **Learning rate:** $5 \times 10^{-6}$ with cosine annealing
- **Batch configuration:** 2 samples per device, 4 gradient accumulation steps
- **Generation:** $M = 4$ candidate completions per group for advantage estimation (Eq. 3)
- **KL penalty:** $\beta = 0.01$
- **Reward weights:** $w_{\text{acc}} = 2.0$, $w_{\text{fmt}} = 1.0$
- **Token budget:** `max_completion_length` = 5120 tokens for $N = 3$ problems
- **Qwen3-4B token budget:** `max_completion_length` = 8000 (other reward and optimization settings unchanged unless otherwise stated)
- **Precision:** Mixed-precision (bfloat16) on NVIDIA RTX PRO 6000 GPUs

**Token Budget Design.** The token budget of 5120 for $N=3$ problems corresponds to approximately 1,707 tokens per problem—roughly one-third of the baseline's average usage (5,000+ tokens). This aggressive budget is intentional: it forces the model to discover compression strategies early in training rather than gradually. For each base model, we select $B_{\max}$ by first estimating the average output length on a small probe set of questions ($L_{\text{avg}}$), then applying a target compression ratio $\lambda$ (default $\lambda \approx 0.5$):

$$B_{\max} \approx N \times L_{\text{avg}} \times \lambda.$$

Algorithm 1 summarizes the procedure. For the JustRL-1.5B setting with $N=3$, this estimate leads to the default budget of 5120. The main-text ablation in §B.3 compares 4096, 5120, and 6144 and shows a clear trade-off: 4096 is too restrictive and hurts accuracy, while 6144 relaxes the budget pressure and weakens token-efficiency gains. Unless otherwise specified, the analyses in this paper use the JustRL-1.5B run with 5120 tokens; the Qwen3-4B configuration follows the same procedure with an 8000-token budget.

---

**Algorithm 1** Selecting the BCR token budget $B_{\max}$

---

**Require:** Base model $\pi_{\text{base}}$, probe set $\mathcal{Q}_{\text{probe}}$, group size $N$, target compression ratio $\lambda$
**Ensure:** Training budget $B_{\max}$
 1: **for** each question $q \in \mathcal{Q}_{\text{probe}}$ **do**
 2:     Generate a completion $y_q \sim \pi_{\text{base}}(\cdot \mid q)$
 3:     Record the output length $\ell_q \leftarrow |y_q|$
 4: **end for**
 5: $L_{\text{avg}} \leftarrow \frac{1}{|\mathcal{Q}_{\text{probe}}|} \sum_{q \in \mathcal{Q}_{\text{probe}}} \ell_q$
 6: $\widetilde{B} \leftarrow N \times L_{\text{avg}} \times \lambda$
 7: $B_{\max} \leftarrow$ nearest practical token cap to $\widetilde{B}$
 8: **return** $B_{\max}$

---

**Data Construction.** Our training data comprises 3,000 groups constructed from DeepMath-103K (He et al., 2025b). For each group, we use stratified sampling based on estimated problem difficulty. Difficulty is proxied by the base model's average reasoning length on each problem (computed from 4 rollouts): problems requiring more tokens are considered harder. Each group is constructed to have approximately the same average difficulty, ensuring that the model encounters a balanced mix of easy and hard problems in every training example. Within each group, problem order is randomized to prevent the model from learning position-dependent strategies (e.g., always allocating more tokens to the last problem).

**Training Infrastructure.** All experiments were conducted on a single node with $4\times$ NVIDIA RTX PRO 6000 GPUs (48GB VRAM each). **Training time is a key practical result:** for the JustRL-1.5B setting, the full 3-epoch run over 1,000 groups with $M=4$ rollouts takes approximately **50 hours** wall-clock. This corresponds to about **200 GPU-hours**, computed as $4 \times 50$ (number of GPUs $\times$ wall-clock hours). For BCR-Qwen3-4B-Thinking-2507, the corresponding run takes approximately **180 hours** wall-clock on the same 4-GPU node, i.e., about **720 GPU-hours** ($4 \times 180$). Despite the increased runtime for the larger backbone, both settings remain single-stage, single-node training jobs and still avoid multi-stage scheduling overhead. This remains substantially lighter than pipelines such as ProRL (8 stages) or FastCuRL (5 stages), which require multiple sequential runs with different configurations.

## A.2. Answer Extraction Algorithm

Reliable answer extraction from multi-problem completions is non-trivial. Mathematical expressions frequently contain nested braces (e.g., \boxed{\dfrac{1}{5}}), which naive regex-based approaches fail to parse correctly. We implement a stack-based parser (Algorithms 2–3) with a three-stage fallback strategy.

---

**Algorithm 2** Stack-based Answer Extraction

---

**Require:** Completion text $y$, number of problems $N$
**Ensure:** Extracted answers $[\hat{a}_1, \ldots, \hat{a}_N]$
1: answers $\leftarrow$ [None] $\times N$
2: sections $\leftarrow$ split$(y,$ "### Problem")
3: **for** each section $s$ with number $i \in [1, N]$ **do**
4:     pattern $\leftarrow$ "Answer$i$: \boxed{"
5:     match $\leftarrow$ search(pattern, $s$)
6:     **if** match found **then**
7:         pos $\leftarrow$ match.end()
8:         answers$[i-1] \leftarrow$ extractBoxed$(s, $pos$)$ {Call Alg. 3}
9:     **end if**
10: **end for**
11: {Fallback: position-based ordering of all boxed expressions}
12: **for** $i \leftarrow 1$ to $N$ **do**
13:     **if** answers$[i-1]$ = None **then**
14:         all_boxed $\leftarrow$ findAllBoxed$(y)$
15:         **if** $i \leq |$all_boxed$|$ **then**
16:             answers$[i-1] \leftarrow$ all_boxed$[i-1]$
17:         **end if**
18:     **end if**
19: **end for**
20: **return** answers

---

**Algorithm 3** Stack-based Boxed Content Parser

---

**Require:** Text $y$, start position pos (first character after opening brace)
**Ensure:** Content inside \boxed{...} or None
1: stack $\leftarrow 1$ {Already past opening brace}
2: $i \leftarrow$ pos
3: **while** $i < |y|$ **and** stack $> 0$ **do**
4:     **if** $y[i] =$ "{" **then**
5:         stack $\leftarrow$ stack $+ 1$
6:     **else if** $y[i] =$ "}" **then**
7:         stack $\leftarrow$ stack $- 1$
8:         **if** stack $= 0$ **then**
9:             **return** $y[$pos $: i]$
10:         **end if**
11:     **else if** $y[i] =$ "\" **and** $i + 1 < |y|$ **then**
12:         $i \leftarrow i + 1$ {Skip escaped character}
13:     **end if**
14:     $i \leftarrow i + 1$
15: **end while**
16: **return** None

---

**Three-Stage Fallback Strategy.** The extraction pipeline employs three stages of decreasing specificity to maximize answer recovery:

*Stage 1: Section-specific pattern matching.* The primary strategy splits the completion by `### Problem` headers and searches for the exact pattern "Answer$i$: `\boxed{...}`" within each section. This is the most reliable strategy and succeeds on >95% of completions. The stack-based parser correctly handles arbitrarily nested braces, ensuring that expressions like $\boxed{\dfrac{a^2 + b^2}{c \cdot (d + e)}}$ are extracted in full.

*Stage 2: Global pattern search.* When section headers are missing or malformed (which occurs more frequently under tight token budgets where the model may skip formatting), the parser searches for all "Answer$i$:" patterns globally in the completion text, regardless of section boundaries. This stage recovers answers from approximately 3% of completions.

*Stage 3: Position-based ordering.* As a last resort, the parser collects all `\boxed{...}` expressions in the completion and assigns them to problems by position (first boxed expression $\rightarrow$ Problem 1, etc.). This heuristic is imperfect—it can fail when models produce intermediate boxed expressions during reasoning—but it recovers answers from approximately 1% of completions that would otherwise yield no output.

**Verification Pipeline.** After extraction, each candidate answer $\hat{a}_i$ is verified against the ground truth $a_i$ using a multi-stage verification process: (1) symbolic equivalence checking via SymPy when both expressions are parseable, (2) numeric comparison with tolerance $\epsilon = 10^{-6}$ for numeric answers, and (3) string matching after LaTeX normalization (removing whitespace, standardizing fraction notation, etc.) as a final fallback. This multi-stage verification ensures that mathematically equivalent but syntactically different answers (e.g., $\frac{1}{2}$ vs. $0.5$ vs. $\frac{2}{4}$) are correctly recognized.

**Extraction Reliability.** Across all training runs and evaluations, our extraction pipeline achieves a 99.2% answer recovery rate (i.e., successfully extracting a parseable answer from the completion). The remaining 0.8% of failures are predominantly caused by severely truncated completions where the model runs out of tokens before producing any boxed answer for the last problem in a group. These cases receive zero accuracy reward, which naturally incentivizes the model to budget tokens appropriately.

# B. Extended Experimental Results

## B.1. Complete Group Size Ablation

Table 5 extends the group size ablation from the main paper (Table 4) to the remaining two benchmarks: MATH-500 and Olympiad. Combined with the main paper results, this provides a comprehensive view of how training group size affects the learned efficiency policy across all five benchmarks.

*Table 5.* Group size ablation on MATH-500 and Olympiad (extending Table 4). All models trained for 300 steps with the specified $N$ and evaluated at $N=1$.

| $N$ | MATH-500 | | Olympiad | |
|---|---|---|---|---|
| | acc. (%) | tokens | acc. (%) | tokens |
| 3 | **90.0** | 2185 | 61.4 | 4052 |
| 4 | 89.4 | 2386 | 62.0 | 4594 |
| 5 | 89.2 | **2001** | **64.4** | **3873** |

**Detailed Analysis.** The results on these two additional benchmarks complement and nuance the findings from the main paper. We organize our analysis around four key observations.

*(1) Difficulty-dependent optimal group size.* The most striking finding is that the optimal $N$ depends on benchmark difficulty. On MATH-500—a benchmark where the baseline already achieves 91.4% accuracy, indicating relatively moderate difficulty—$N=3$ yields the best accuracy (90.0%). However, on Olympiad—the most challenging benchmark in our suite, where the baseline achieves only 62.1%—$N=5$ achieves the best accuracy (64.4%) *and* the lowest token count (3,873). This counterintuitive result suggests that for harder problems, stronger implicit pressure from larger $N$ may actually *help* the model focus on essential reasoning steps by more aggressively pruning the verbose exploration that characterizes baseline models on difficult tasks.

*(2) Accuracy stability across configurations.* Across all five benchmarks (combining Table 4 and Table 5), accuracy varies by less than 6% across group sizes. On MATH-500, the range is 89.2–90.0%; on Olympiad, it is 61.4–64.4%. This robustness is practically important: it means that BCR's benefits are not critically sensitive to the choice of $N$, and practitioners can select $N$ based on infrastructure constraints without risking significant accuracy degradation.

*(3) Token efficiency patterns.* Token usage shows more variation across $N$ than accuracy, but the patterns are consistent. On MATH-500, $N=5$ achieves the lowest token count (2,001) followed by $N=3$ (2,185) and $N=4$ (2,386). The non-monotonic relationship between $N$ and token usage on MATH-500 (where $N=4$ uses *more* tokens than $N=3$) may reflect the interaction between optimization difficulty and compression pressure: at $N=4$, the model faces harder optimization than $N=3$ but less aggressive pressure than $N=5$, potentially leading to suboptimal convergence in 300 training steps. On Olympiad, the pattern is also non-monotonic, with $N=4$ yielding the highest token count (4,594). This suggests that training for more steps could resolve these non-monotonicities.

*(4) Comparison with baseline.* To contextualize these results, recall that the JustRL baseline uses 3,099 tokens on MATH-500 and 7,017 tokens on Olympiad. All BCR configurations achieve substantial reductions: even the least efficient configuration ($N=4$ on MATH-500) uses only 2,386 tokens—a 23% reduction. The most efficient ($N=5$ on MATH-500) achieves a 35% reduction. On Olympiad, reductions range from 35% ($N=4$) to 45% ($N=3$). These gains are achieved with <2% accuracy loss on MATH-500 and with accuracy *improvements* on Olympiad, reinforcing the "free lunch" phenomenon described in the main paper.

**Recommendation.** Based on the comprehensive five-benchmark analysis, we recommend $N=3$ as the default group size for general use. It provides the most consistent accuracy-efficiency trade-off, avoids the optimization challenges of larger $N$, and achieves strong compression on both easy and hard benchmarks. For deployment scenarios that prioritize maximum efficiency on challenging problems, $N=5$ may be preferred, but with the caveat that extended training may be necessary to fully realize its potential.

## B.2. Complete Multi-Problem Inference Results

Table 6 extends the task-scaling analysis from the main paper (Table 2) to the MATH-500 and Olympiad benchmarks, providing a complete picture of how both models behave under varying inference-time concurrency.

*Table 6.* Task-scaling law on MATH-500 and Olympiad (extending Table 2). Both models evaluated under $N\times$ concurrent inference. Bold indicates best within each $N$.

| $N$ | Model | MATH-500 | | Olympiad | |
|---|---|---|---|---|---|
| | | acc. (%) | tokens | acc. (%) | tokens |
| $1\times$ | JustRL-1.5B | **91.4** | 3099 | 62.1 | 7017 |
| | BCR (ours) | 87.6 | **1868** | **62.9** | **2969** |
| $2\times$ | JustRL-1.5B | 83.0 | 2033 | 47.3 | 5567 |
| | BCR (ours) | **85.6** | **1290** | **52.0** | **1919** |
| $3\times$ | JustRL-1.5B | 73.6 | 2544 | 38.7 | 4696 |
| | BCR (ours) | **82.4** | **1063** | **45.7** | **1528** |
| $4\times$ | JustRL-1.5B | 65.8 | 1957 | 31.6 | 3902 |
| | BCR (ours) | **79.6** | **878** | **41.7** | **1148** |
| $5\times$ | JustRL-1.5B | 63.2 | 2205 | 26.1 | 3222 |
| | BCR (ours) | **79.0** | **839** | **40.0** | **1051** |

**Detailed Analysis.** The MATH-500 and Olympiad results confirm and strengthen the task-scaling law described in the main paper. We present a comprehensive analysis organized around five key findings.

*(1) BCR dominates across all $N$ on both benchmarks.* Starting from $N=2$, BCR achieves both higher accuracy and lower token usage on every benchmark. At $N=1$, BCR trails the baseline by 3.8% on MATH-500 accuracy (87.6% vs. 91.4%) but compensates with a 40% token reduction (1,868 vs. 3,099). On Olympiad, BCR leads on both metrics even at $N=1$ (62.9% vs. 62.1% accuracy, with 58% fewer tokens). This demonstrates that the efficiency learned during batched training transfers fully to standard single-problem inference.

*(2) The accuracy gap widens dramatically with $N$.* This is the most compelling evidence for BCR's adaptive resource allocation capability. On MATH-500, as $N$ increases from 1 to 5:

- The baseline drops from 91.4% to 63.2%—a 31% relative decline.
- BCR drops from 87.6% to 79.0%—only a 10% relative decline.

On Olympiad, the contrast is even more dramatic:

- The baseline drops from 62.1% to 26.1%—a 58% relative decline.
- BCR drops from 62.9% to 40.0%—only a 36% relative decline.

The baseline's accuracy on Olympiad at $N=5$ (26.1%) is barely above random guessing on many competition problems, while BCR retains 40.0%—demonstrating that BCR trains the model to allocate resources adaptively under pressure rather than simply truncating reasoning uniformly.

*(3) Token efficiency follows a smooth, predictable curve for BCR.* An important practical finding is that BCR's per-problem token usage decreases monotonically with $N$ on both benchmarks: $1{,}868 \rightarrow 1{,}290 \rightarrow 1{,}063 \rightarrow 878 \rightarrow 839$ on MATH-500, and $2{,}969 \rightarrow 1{,}919 \rightarrow 1{,}528 \rightarrow 1{,}148 \rightarrow 1{,}051$ on Olympiad. In contrast, the baseline's token usage is erratic—on Olympiad, it decreases from $N=1$ to $N=4$ ($7{,}017 \rightarrow 3{,}902$) but then *increases* at $N=5$ (3,222), and on MATH-500 it similarly shows non-monotonic behavior. BCR's smooth scaling curve means that $N$ can be reliably used as a throughput-accuracy knob in deployment: doubling $N$ from 1 to 2 roughly halves per-problem cost while maintaining accuracy, with further diminishing returns at higher $N$.

*(4) BCR achieves greater absolute accuracy gains on harder benchmarks.* At $N=3$, BCR outperforms the baseline by 8.8% on MATH-500 (82.4% vs. 73.6%) and by 7.0% on Olympiad (45.7% vs. 38.7%). At $N=5$, the gaps widen to 15.8%

on MATH-500 and 13.9% on Olympiad. This pattern—larger absolute accuracy advantages under higher concurrency—suggests that BCR's learned compression strategies become increasingly valuable as resource constraints tighten. In practical deployment scenarios where batching is used to improve throughput, BCR offers substantial accuracy advantages over baseline models.

*(5) Implications for cost-efficient deployment.* Combining the efficiency and accuracy results, BCR enables favorable cost-accuracy trade-offs. For example, a practitioner could solve 5 problems per API call using BCR at $N{=}5$ and achieve 79.0% accuracy on MATH-500 with only 839 tokens per problem. The baseline at $N{=}1$ achieves 91.4% accuracy but requires 3,099 tokens—a $3.7\times$ higher per-problem cost. For applications where 79% accuracy is acceptable, BCR at $N{=}5$ provides a $3.7\times$ cost reduction while processing $5\times$ more problems per call—an $18.5\times$ improvement in cost-adjusted throughput.

## B.3. Complete Token Budget Ablation Results

Table 7 consolidates the complete token-budget ablation across all five benchmarks, reporting accuracy (%) and average generated tokens per problem in the same grouped format as the main tables.

*Table 7.* Complete token budget ablation across all five benchmarks. The 1.5B model is trained with $N=3$. We report accuracy (%) and average generated tokens per problem.

| $B_{\max}$ | AIME25 | | AMC23 | | Minerva | | MATH-500 | | Olympiad | |
| | acc. (%) | tokens | acc. (%) | tokens | acc. (%) | tokens | acc. (%) | tokens | acc. (%) | tokens |
|---|---|---|---|---|---|---|---|---|---|---|
| 4096 | 26.7 | 2561 | 82.5 | 2135 | 46.0 | 2085 | 86.6 | 1524 | 61.1 | 2711 |
| **5120 (Ours)** | 33.3 | 3173 | 87.5 | 2637 | 48.5 | 2494 | 87.6 | 1868 | 62.9 | 2969 |
| 6144 | 30.0 | 3527 | 80.0 | 2701 | 44.9 | 2875 | 88.8 | 1874 | 63.4 | 3116 |

**Detailed Analysis.** The MATH-500 and Olympiad columns refine the budget-selection story from the main paper rather than changing it. We summarize the main takeaways below.

*(1) 4096 is consistently too restrictive.* On both benchmarks, the most aggressive budget yields the lowest token usage, but this comes with the weakest accuracy: 86.6% on MATH-500 and 61.1% on Olympiad. This confirms the same pattern seen on AIME25, AMC23, and Minerva in the main paper: when the budget is too tight, the model begins sacrificing completeness for compression, which directly hurts final-answer quality.

*(2) 6144 slightly improves accuracy on these two benchmarks, but only at higher token cost.* Relative to 5120, the looser 6144 budget improves MATH-500 from 87.6% to 88.8% and Olympiad from 62.9% to 63.4%. However, it also increases token usage from 1,868 to 1,874 on MATH-500 and from 2,969 to 3,116 on Olympiad. The gain is therefore modest in accuracy and negative in efficiency, especially on Olympiad where the per-problem token cost rises by 147 tokens.

*(3) The cross-benchmark optimum is still 5120.* The key point is that these two benchmarks should be interpreted jointly with the main-paper results. On AIME25, AMC23, and Minerva, 5120 clearly outperforms 6144. On MATH-500 and Olympiad, 6144 recovers a small amount of extra accuracy, but not enough to offset its weaker efficiency and its regressions on the other three benchmarks. This is exactly the behavior we want from a default hyperparameter: 5120 is not always the single best point on every benchmark, but it is the most reliable setting across the full evaluation suite.

*(4) Harder benchmarks benefit somewhat more from relaxed budgets.* Compared with AMC23 and Minerva, Olympiad is more tolerant of a looser budget, which is consistent with its longer and less forgiving reasoning chains. This suggests a practical deployment interpretation: if one were optimizing only for a harder benchmark family and cared less about token efficiency, a slightly larger budget could be reasonable. For the paper's main goal—a single efficient setting with strong performance across diverse benchmarks—5120 remains the better choice.

**Recommendation.** We retain $B_{\max}=5120$ as the default. It is the best overall compromise between compression pressure and retained accuracy across all five benchmarks, while 4096 is too restrictive and 6144 is too permissive for the general setting considered in this work.

## B.4. Extended Efficiency-Accuracy Pareto Frontiers

Building upon the efficiency analysis presented in the main text, Figure 5 visualizes the training trajectories and the resulting Pareto frontiers for the remaining four mathematical reasoning benchmarks: AIME25, AMC23, MATH-500, and Olympiad.

The scatter plots map the average token usage (x-axis, lower is better) against task accuracy (y-axis, higher is better). For both model families, we plot the intermediate checkpoints during BCR training, indicated by the connected trajectories with arrows. The final converged models are marked with prominent stars (orange for **BCR-JustRL-1.5B**, blue for **BCR-Qwen3-4B**). For context, we also plot various efficiency-agnostic and length-controlled baseline models as isolated points.

Across all four benchmarks, the training dynamics exhibit a consistent and highly stable pattern: BCR optimization systematically pushes the models toward the top-left quadrant. The arrows demonstrate that the models successfully learn to compress their reasoning traces (moving left) while strictly preserving or even enhancing their problem-solving capabilities (moving up or remaining stable). The final BCR models establish a superior Pareto frontier compared to existing methods, confirming that the implicit budget constraint effectively crystallizes latent reasoning efficiency without the severe accuracy degradation often associated with naive length-minimization approaches.

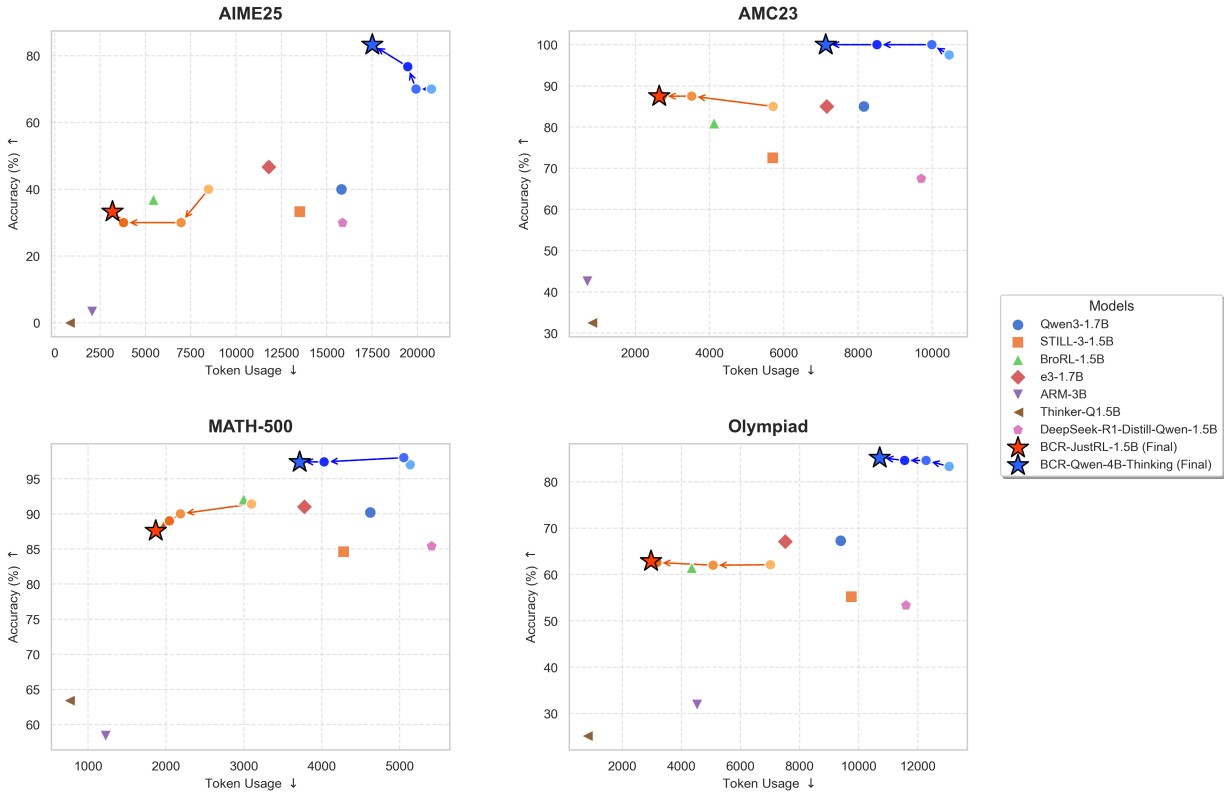

*Figure 5.* **Extended Efficiency-Accuracy Pareto Frontiers.** Training trajectories on AIME25, AMC23, MATH-500, and Olympiad. The arrows track intermediate model checkpoints during BCR optimization, demonstrating a continuous, stable shift toward lower token consumption and competitive accuracy. The final BCR models (stars) consistently dominate the Pareto frontier relative to baseline models of comparable or larger scale.

# C. Qualitative Analysis

## C.1. Overview and Methodology

To understand how BCR achieves token efficiency without sacrificing reasoning quality, we present detailed case studies from AIME 2025 (§C.2.1), AMC 2023 (§C.2.2), and BRUMO 2025 (§C.2.3). Our qualitative analysis examines seven representative problems selected to illustrate the diversity of BCR's compression mechanisms.

**Selection Methodology.** We selected examples according to three criteria: (1) *diversity of mathematical domains*—spanning algebra, geometry, number theory, complex analysis, and arithmetic; (2) *diversity of compression mechanisms*—covering metacognitive loop elimination, strategy exploration reduction, pedagogical narration removal, and catastrophic degeneration prevention; and (3) *diversity of outcomes*—including cases where both models succeed (to isolate style differences), cases where BCR succeeds and the baseline fails (to illustrate accuracy benefits), and cases with varying degrees of token reduction (37–92%).

**Comparison Protocol.** For each example, we show excerpts from both models' reasoning traces, highlighting key differences. We use **green** to mark efficient, information-dense reasoning in BCR outputs and *red* to mark verbose or failed patterns in baseline outputs. Token counts are measured as total generated tokens (excluding the prompt) for each model's complete response on the given problem.

**Taxonomy of Compression Mechanisms.** Our analysis reveals four systematic compression mechanisms that BCR induces:

1. **Metacognitive Loop Elimination.** The baseline model frequently interrupts correct reasoning with self-verification phrases: "*Wait, wait, wait. . .*", "*let me check if this makes sense*", "*let me double-check. . .*" These loops consume 200–500 tokens per instance without contributing new mathematical content—they revisit calculations that were already correct. BCR-trained models proceed linearly: once a valid approach is identified, reasoning flows to the conclusion without unnecessary self-doubt. Importantly, BCR models still verify edge cases and boundary conditions when *mathematically necessary* (e.g., checking that solutions fall within specified ranges); they simply avoid *redundant* verification of already-correct intermediate results.

2. **Strategy Exploration Reduction.** Baseline models often explore multiple solution strategies before committing to one, even when the first approach is correct. For instance, on a factorization problem, the baseline might attempt both polynomial long division and the Remainder Theorem before selecting one. BCR models apply the most direct strategy immediately, suggesting that batched training improves problem recognition—the model learns to identify the most efficient solution path faster, likely because time spent on suboptimal strategies for one problem directly reduces the budget available for subsequent problems.

3. **Pedagogical Narration Removal.** Baseline models sometimes explain basic mathematical concepts (e.g., "when you raise a power to a power, you multiply the exponents") that are unnecessary given the model's demonstrated competence. This "teaching mode" behavior likely arises from instruction-tuning data that includes step-by-step explanations. BCR suppresses this tendency because pedagogical narration consumes tokens without improving accuracy.

4. **Catastrophic Degeneration Prevention.** On hard problems requiring long reasoning chains, baseline models sometimes exhaust their 32K token budget and degenerate into repetitive, non-mathematical character sequences. BCR's implicit budget constraint trains the model to commit to an answer before entering these degenerate generation regimes, effectively preventing a failure mode that the baseline is susceptible to.

Crucially, all four mechanisms represent *syntactic* rather than *semantic* compression. BCR does not teach models to skip reasoning steps; it teaches them to execute each step once and move forward. This distinction is critical: it explains why BCR can achieve large token reductions without sacrificing—and sometimes improving—accuracy.

**C.2. BCR-JustRL-1.5B Case Studies**

C.2.1. AIME 2025 EXAMPLES

We present three AIME 2025 examples spanning algebra, geometry, and number theory. AIME problems are competition-level and require multi-step mathematical reasoning, making them ideal for studying the quality of BCR's compression.

**Example 1 (Table 8): Equation Factorization (49.5% reduction).** This example best illustrates *metacognitive verbosity*. The problem asks for ordered pairs $(x, y)$ satisfying $12x^2 - xy - 6y^2 = 0$. Both models correctly factorize this as $(3x + 2y)(4x - 3y) = 0$ and identify the two solution families. However, their reasoning diverges sharply after this point.

The baseline model, having correctly identified that $y$ must be divisible by 4 for the first family and by 3 for the second, enters a prolonged self-interruption spiral. The phrase "*Wait wait wait*" appears multiple times, and the model re-derives the divisibility constraints, re-counts the solutions, and even temporarily arrives at an incorrect intermediate answer ($N = 2016$) before correcting itself. This metacognitive churning consumes over 3,500 tokens—nearly half the total output—without introducing any new mathematical content.

The BCR model, in contrast, executes a clean four-step solution: (1) factor the equation, (2) enumerate the integer solutions for each family, (3) identify the overlap at $(0, 0)$, and (4) apply inclusion-exclusion. Each step is executed once. The 49.5% token reduction comes entirely from eliminating redundant verification, not from shortcutting any mathematical reasoning.

**Example 2 (Table 9): Coordinate Geometry (37.6% reduction).** This problem involves computing the area of a heptagon using the shoelace formula. Both models correctly set up and compute the formula. The key difference is in post-computation behavior: the baseline model, having obtained the correct answer of 588, immediately questions its own result ("*Wait, but let me check if this makes sense*") and proceeds to re-verify individual terms of the shoelace formula. It revisits term6 and term7 explicitly, confirming values it had already computed correctly.

The BCR model lists all seven terms in a single enumeration, computes the sum in one line, and terminates. This pattern—*compute once, trust the result*—is a hallmark of BCR-trained models. The moderate 37.6% reduction reflects that the baseline's redundancy on this geometric problem is less severe than on Example 1, as the shoelace computation is more mechanical and offers fewer opportunities for self-doubt spirals.

**Example 3 (Table 10): Divisibility Analysis (37.2% reduction).** This example highlights *strategy exploration reduction*. The problem asks for integers $n$ such that $n + 2$ divides $3(n + 3)(n^2 + 9)$. The baseline begins with informal deliberation ("*Hmm, let us break this down step by step*"), considers using polynomial division, attempts to simplify the expression, then discovers—after considerable exploration—that the Remainder Theorem provides the cleanest approach. Only after this exploration does it compute the remainder (39) and enumerate the divisors.

The BCR model skips this exploration entirely. It opens with "The condition is that $n + 2$ divides $3(n + 3)(n^2 + 9)$" and immediately applies the Remainder Theorem. Within five lines, it has the answer. The 37.2% reduction reflects the elimination of the exploratory preamble. This example suggests that BCR training improves *problem recognition*: the model learns to identify the most efficient solution strategy faster, likely because wasting tokens on suboptimal strategies for one problem reduces the budget available for subsequent problems in the training batch.

*Table 8.* **AIME 2025, Example 1: Equation Factorization** (49.5% token reduction). Both models reach the correct answer via factorization, but the baseline wastes >3,500 tokens on self-interruption loops that revisit already-correct calculations.

| JustRL-DeepSeek-1.5B (7,211 tokens) | BCR-JustRL-1.5B (3,639 tokens) |
|---|---|
| **Problem:** Find the number of ordered pairs $(x, y)$ with $x, y \in [-100, 100]$ such that $12x^2 - xy - 6y^2 = 0$. **Ground Truth:** 117 | |
| So the solutions are $x = (3y)/4$ or $x = (-2y)/3$. But since $x$ and $y$ are integers, these expressions must result in integers. So for $x = (3y)/4$ to be integer, $3y$ must be divisible by 4, which implies that $y$ must be divisible by 4, since 3 and 4 are coprime. Similarly... *Wait wait wait, wait. Wait wait, wait wait...* *[Extensive rechecking of calculations]* *Wait the final answer is $N = 2016$... Wait wait wait let me double-check...* | The equation factors as $(3x + 2y)(4x - 3y) = 0$. **Therefore, either** $3x + 2y = 0$ **or** $4x - 3y = 0.$ For $3x + 2y = 0$: $x = 2k$, $y = -3k$, with $k \in [-33, 33]$. This gives 67 solutions. For $4x - 3y = 0$: $x = 3k$, $y = 4k$, with $k \in [-25, 25]$. This gives 51 solutions. **Overlap is only** $(0, 0)$**, so total** $= 67 + 51 - 1 = 117.$ |
| **Answer:** $\boxed{117}$ ✓ | **Answer:** $\boxed{117}$ ✓ |

*Table 9.* **AIME 2025, Example 2: Coordinate Geometry** (37.6% token reduction). The baseline re-verifies each shoelace term individually despite having computed them correctly; BCR computes once and terminates.

| JustRL-DeepSeek-1.5B (6,676 tokens) | BCR-JustRL-1.5B (4,164 tokens) |
|---|---|
| **Problem:** In $\triangle ABC$, points on sides with given lengths. $M, N$ are reflections. Area of $DEGF$ is 288. Find area of heptagon $AFNBCEM$. **Ground Truth:** 588 | |
| The shoelace formula applied to the heptagon gave $\frac{1}{2} \times 1176 = 588$. *So the area is 588. Wait, but let me check if this makes sense.* *Given that the area of triangle ABC is 588, and the heptagon AFNBCEM is covering most of it, but the calculation gives 588.* *Wait, but let me think again.* *But wait, let me check the coordinates again. Wait, when I calculated term6... Yes, that is correct. And term7... So the sum is indeed...* | Now compute the sum: Term1=0, Term2=$-240$, Term3=$+840$, Term4=$+1176$, Term5=$-840$, Term6=$+240$, Term7=0. **Total sum** $= 0-240+840+1176-840+240+0 = 1176.$ Area $= \frac{1}{2} \times |1176| = 588$. **So the area of the heptagon is 588.** |
| **Answer:** $\boxed{588}$ ✓ | **Answer:** $\boxed{588}$ ✓ |

*Table 10.* **AIME 2025, Example 3: Divisibility Analysis** (37.2% token reduction). The baseline explores multiple strategies before finding the Remainder Theorem approach; BCR applies it directly without exploration.

| JustRL-DeepSeek-1.5B (3,530 tokens) | BCR-JustRL-1.5B (2,216 tokens) |
|---|---|
| **Problem:** Find the sum of all positive integers $n$ such that $n + 2$ divides $3(n + 3)(n^2 + 9)$. **Ground Truth:** 49 | |
| Okay, so I have got this math problem here... *Hmm, let us break this down step by step.* First, let us write down what is given... *To tackle this, maybe I can use polynomial division or factor the expression somehow. Let me try to see if I can simplify...* Alternatively, since $n+2$ divides the product, by the Factor Theorem, if we substitute $n = -2$... *Wait, but actually, if $n+2$ divides the product, then substituting $n = -2$ should give zero. Let us test that.* *[Performs explicit polynomial long division]* So the remainder is 39. Therefore... 39 must be divisible by $(n + 2)$. | The condition is that $n + 2$ divides $3(n + 3)(n^2 + 9)$. **Using the Remainder Theorem:** substituting $n = -2$ into the numerator gives $3(1)(13) = 39$. **So $n + 2$ must be a divisor of 39.** Divisors of 39: 1, 3, 13, 39. Since $n$ is positive, $n + 2 \geq 3$, so $n \in \{1, 11, 37\}$. Sum: $1 + 11 + 37 = 49$. |
| **Answer:** $\boxed{49}$ ✓ | **Answer:** $\boxed{49}$ ✓ |

C.2.2. AMC 2023 EXAMPLES

Tables 11–13 present three AMC 2023 examples spanning complex analysis, arithmetic, and geometry. AMC problems are generally easier than AIME, which allows us to observe how BCR handles problems of varying difficulty.

**Example 4 (Table 11): Complex Analysis (67.8% reduction).** This is the most striking example in our entire analysis, demonstrating that BCR's efficiency can also lead to *higher accuracy*—a direct illustration of the "free lunch" phenomenon reported in the main results. The problem asks for the maximum imaginary part of $z$ satisfying $|1 + z + z^2| = 4$.

The baseline model generates over 10,000 tokens of verbose exploration but arrives at an *incorrect answer* ($m + n = 5$). Post-hoc analysis reveals the failure mechanism: the model's extended deliberation leads it through multiple possible approaches, and in the process it confuses the constraint satisfaction conditions. Specifically, it considers a candidate $y = \sqrt{3}/2$ but incorrectly evaluates whether the constraint $|1 + z + z^2| = 4$ is satisfied at this point, ultimately concluding with the wrong values $m = 3, n = 2$.

The BCR model, by contrast, identifies the key algebraic insight immediately: to maximize Im$(z)$, set $1 + z + z^2 = -4$ (the point on the circle $|w| = 4$ that is farthest from the real axis when mapped back). This yields the quadratic $z^2 + z + 5 = 0$, whose discriminant gives Im$(z) = \sqrt{19}/2$, producing the correct answer $m + n = 19 + 2 = 21$. The entire solution takes only 3,261 tokens.

This example supports a key hypothesis: by eliminating unproductive deliberation loops, BCR reduces the probability of the model "reasoning itself into an error." When the baseline explores multiple approaches, it creates more opportunities for confusion and incorrect intermediate conclusions. BCR's implicit budget pressure forces the model to commit to the most promising strategy early, which paradoxically leads to more reliable reasoning.

**Example 5 (Table 12): Digit Counting (58.9% reduction).** This simpler problem asks for the number of digits in $8^5 \cdot 5^{10} \cdot 15^5$. The baseline's verbosity here takes a different form: rather than self-doubt, it engages in *pedagogical narration*, explaining basic mathematical concepts. For instance, it states "*When you raise a power to a power, you multiply the exponents*"—a rule that the model clearly already knows and applies correctly. This "teaching mode" behavior likely arises from instruction-tuning data that contains step-by-step explanations designed for human learners. BCR suppresses this tendency because pedagogical tokens consume budget without improving accuracy.

The BCR model produces a concise algebraic chain: $8^5 = 2^{15}$, $15^5 = 3^5 \cdot 5^5$, combine to get $10^{15} \cdot 243$, count digits: $3 + 15 = 18$. The entire reasoning fits in 1,425 tokens—less than half the baseline's 3,464. The 58.9% reduction is driven almost entirely by eliminating unnecessary narration, not by skipping any computational steps.

**Example 6 (Table 13): Triangle Area (49.6% reduction).** This geometry problem requires applying the Law of Cosines to find a triangle's sides, then computing its area. The baseline arrives at the correct answer but wastes tokens on two distinct self-doubt episodes: first, "*Wait, wait, wait a second*" after finding $d = 4$ (despite having correctly derived this from a quadratic), and second, "*Let me verify if these satisfy triangle inequality*" (a valid check, but the triangle inequality is trivially satisfied for sides 6, 10, 14 and could be verified mentally).

The BCR model presents a linear derivation: set up Law of Cosines $\rightarrow$ solve quadratic for $d$ $\rightarrow$ compute area using $\sin 120°$ $\rightarrow$ identify $m$ and $n$ $\rightarrow$ answer. Each step follows naturally from the previous one, with no backtracking or re-verification. The resulting 49.6% token reduction illustrates that even on problems where the baseline reasons correctly, BCR produces substantially more concise output by eliminating redundant verification that adds no mathematical value.

*Table 11.* **AMC 2023, Example 4: Complex Analysis** (67.8% token reduction). BCR correct, baseline incorrect. The baseline's verbose exploration leads to a reasoning error through constraint confusion.

| JustRL-DeepSeek-1.5B (10,122 tokens) | BCR-JustRL-1.5B (3,261 tokens) |
|---|---|
| **Problem:** Consider the set of complex numbers $z$ satisfying $|1 + z + z^2| = 4$. The maximum value of the imaginary part of $z$ can be written as $\frac{\sqrt{m}}{n}$. What is $m + n$? **Ground Truth:** 21 | |
| ... So the maximum $y$ occurs when $y = \sqrt{3}/2$ and the condition is not satisfied, so the maximum $y$ is $\sqrt{3}/2$, and the $m + n = 3 + 2 = 5$. *[Incorrect analysis: confuses constraint satisfaction]* *[Concludes with wrong answer]* | Set $1 + z + z^2 = -4$, giving $z^2 + z + 5 = 0$. Solutions: $z = \frac{-1 \pm i\sqrt{19}}{2}$ **Thus $\text{Im}(z) = \pm \frac{\sqrt{19}}{2}$, maximum is $\frac{\sqrt{19}}{2}$.** So $m = 19$, $n = 2$, thus $m + n = 21$. |
| **Answer:** $\boxed{5}$ $\times$ | **Answer:** $\boxed{21}$ $\checkmark$ |

*Table 12.* **AMC 2023, Example 5: Digit Counting** (58.9% token reduction). The baseline narrates basic exponent rules that it already knows; BCR computes directly without unnecessary explanation.

| JustRL-DeepSeek-1.5B (3,464 tokens) | BCR-JustRL-1.5B (1,425 tokens) |
|---|---|
| **Problem:** How many digits are in the base-ten representation of $8^5 \cdot 5^{10} \cdot 15^5$? **Ground Truth:** 18 | |
| Alright, so I have this math problem here... *Hmm, okay. Let me break this down step by step.* Starting with $8^5$. I know that 8 is 2 cubed, so $8 = 2^3$. Therefore, $8^5 = (2^3)^5$. *When you raise a power to a power, you multiply the exponents.* So that would be $2^{15}$... *[Extensive explanation of basic exponent rules]* So $10^{15}$ would have 16 digits. *But let me confirm that.* Yes, $10^n$ is a 1 followed by n zeros... | $8^5 = (2^3)^5 = 2^{15}$ $15^5 = (3 \cdot 5)^5 = 3^5 \cdot 5^5$ **Combined:** $2^{15} \cdot 5^{15} \cdot 3^5 = 10^{15} \cdot 243$ $243 \times 10^{15}$ is 243 followed by 15 zeros. **Digits:** $3 + 15 = 18$ |
| **Answer:** $\boxed{18}$ $\checkmark$ | **Answer:** $\boxed{18}$ $\checkmark$ |

*Table 13.* **AMC 2023, Example 6: Triangle Area** (49.6% token reduction). The baseline rechecks valid intermediate results with two distinct self-doubt episodes; BCR follows a linear derivation from setup to answer.

| JustRL-DeepSeek-1.5B (4,329 tokens) | BCR-JustRL-1.5B (2,182 tokens) |
|---|---|
| **Problem:** Triangle $ABC$ has side lengths in arithmetic progression, smallest side 6, and an angle of $120°$. Find the area in form $m\sqrt{n}$. What is $m + n$? **Ground Truth:** 18 | |
| ... So $d = 4$, sides are $6, 10, 14$. *Wait, wait, wait a second. Wait, the sides are 6, 6 + d, 6 + 2d. If d=4, then sides are 6,10,14.* *Let me verify if these satisfy triangle inequality...* *Wait, wait, wait, wait. Let me double-check the sides.* Area $= \frac{1}{2} \cdot 6 \cdot 10 \cdot \sin(120°) = 15\sqrt{3}$ | Sides: $a = 6$, $b = 6 + d$, $c = 6 + 2d$ Law of Cosines with $\cos(120°) = -\frac{1}{2}$: $(6 + 2d)^2 = 36 + (6 + d)^2 + 6(6 + d)$ **Solving:** $3d^2 + 6d - 72 = 0 \Rightarrow d = 4$ Sides: $6, 10, 14$. Area $= \frac{1}{2}(6)(10)\sin 120° = 15\sqrt{3}$ **So $m = 15$, $n = 3$, $m + n = 18$** |
| **Answer:** $\boxed{18}$ $\checkmark$ | **Answer:** $\boxed{18}$ $\checkmark$ |

C.2.3. BRUMO 2025 EXAMPLE

Table 14 presents the most extreme case in our analysis: the baseline generates over 32,000 tokens and fails entirely, while BCR solves the problem in 2,692 tokens—a 91.8% reduction that also converts a failure into a success.

**Example 7 (Table 14): Roots of Unity (91.8% reduction).** This problem requires finding the smallest positive integer $n$ such that $z^n - 1$ and $(z - \sqrt{3})^n - 1$ share a common complex root. The mathematical insight is that both $\alpha$ and $\alpha - \sqrt{3}$ must be $n$-th roots of unity, and the key is to find a root $\alpha$ on the unit circle such that $\alpha - \sqrt{3}$ also has unit modulus.

The baseline attempts to reason about this systematically but quickly becomes trapped in a repetitive computational loop. It considers various candidate values of $n$, attempts to enumerate roots of unity for each, and frequently backtracks ("*Wait, that doesn't work, let me try...*"). As the reasoning chain grows beyond approximately 15,000 tokens, the model's coherence degrades. Beyond 25,000 tokens, the output degenerates into non-mathematical character sequences: "`2 2 1 11 1 1 1 0, , 11 1`". This degeneration pattern—where extended autoregressive generation causes loss of coherence—is a known failure mode documented in the overthinking literature (Chen et al., 2025a). The model exhausts its full 32,768-token budget without producing any valid answer.

The BCR model avoids this failure entirely through a remarkably direct solution. It identifies the key geometric insight: if $\alpha = e^{i\pi/6}$, then $\alpha$ lies on the unit circle at angle $\pi/6$, and $\alpha - \sqrt{3} = e^{i \cdot 5\pi/6}$ also lies on the unit circle (this can be verified by computing the modulus). Both $e^{i\pi/6}$ and $e^{i \cdot 5\pi/6}$ have order 12 (since $12 \cdot \pi/6 = 2\pi$ and $12 \cdot 5\pi/6 = 10\pi = 5 \cdot 2\pi$). The model verifies both conditions explicitly and terminates, consuming only 2,692 tokens.

This example demonstrates three important properties of BCR:

- **Efficiency prevents catastrophic failure.** The baseline's 32K-token degeneration is not merely an efficiency problem—it is a correctness problem. BCR's implicit budget constraint trains the model to commit to promising approaches rather than endlessly exploring, effectively preventing a class of failure that the baseline is susceptible to.
- **Conciseness correlates with insight.** The BCR model's solution is not just shorter—it demonstrates deeper mathematical insight. Rather than brute-forcing through candidate values of $n$, it identifies the geometric structure of the problem and solves it in a single step. This suggests that BCR training may encourage models to search for elegant solutions rather than exhaustive ones.
- **Order-of-magnitude compression is possible.** The 91.8% reduction ($32{,}677 \rightarrow 2{,}692$ tokens) demonstrates that the gap between verbose and efficient reasoning can be enormous. This extreme case, while not representative of average performance, illustrates the upper bound of BCR's compression potential.

*Table 14.* **BRUMO 2025: Roots of Unity** (91.8% token reduction). The baseline generates 32K tokens and degenerates into non-mathematical character sequences; BCR identifies the key geometric insight and terminates in 2,692 tokens with the correct answer.

| **JustRL-DeepSeek-1.5B (32,677 tokens)** | **BCR-JustRL-1.5B (2,692 tokens)** |
|---|---|
| **Problem:** What is the smallest positive integer $n$ such that $z^n - 1$ and $(z - \sqrt{3})^n - 1$ share a common complex root? | |
| **Ground Truth:** 12 | |
| *[Model generates over 32,000 tokens]* 
 *[Output degenerates into repetitive patterns:]* 
 `... 2 2 1 11 1 1 1 0, , 11 1 , (1 1` 
 `...` 
 *[No valid answer extracted]* | Let $\alpha$ be a common root. Then $\alpha^n = 1$ and $(\alpha - \sqrt{3})^n = 1$. 
 If $\alpha = e^{i\pi/6}$, then $|\alpha| = 1$ and $\alpha - \sqrt{3} = e^{i \cdot 5\pi/6}$. 
 **Order of $e^{i\pi/6}$ is 12. Order of $e^{i \cdot 5\pi/6}$ is also 12.** 
 Verify: $\alpha^{12} = e^{i \cdot 2\pi} = 1$ ✓ 
 $(\alpha - \sqrt{3})^{12} = e^{i \cdot 10\pi} = 1$ ✓ 
 **Smallest such $n$ is 12.** |
| **Answer:** None ✗ | **Answer:** $\boxed{12}$ ✓ |

## C.3. BCR-Qwen-4B Case Studies

### C.3.1. AIME 2025 EXAMPLES

We present three AIME 2025 examples spanning combinatorial enumeration, telescoping algebra, and geometry on the nine-point circle. Compared with the 1.5B case studies in Appendix C.2, the qualitative question for the 4B model is slightly different: Qwen3-4B already solves many difficult problems, so the issue is not simply whether BCR can enforce brevity, but whether it can selectively remove the parts of long-form reasoning that are least coupled to correctness. The three examples below show all of the main regimes. In one case, BCR repairs a missing combinatorial factor and flips the answer from incorrect to correct; in another, both models are correct but BCR compresses away a large amount of interpretive hesitation; and in the third, BCR still reasons at length, but unlike the baseline it commits to a coherent final answer before the trace drifts into unresolved geometric speculation.

**Example 8 (Table 15): Sudoku-Style Counting (43.2% reduction, incorrect → correct).** This is the clearest AIME free-lunch example for the Qwen family. The baseline correctly recognizes the puzzle as a constrained counting problem over row permutations and block structure, but it prematurely compresses the middle-block combinatorics into a much smaller factor. The result is a superficially plausible but incomplete count, which propagates directly into the wrong prime exponents and final score 59. The main token cost comes before the mistake: the model spends a long prefix repeatedly re-describing the grid and block layout instead of isolating the single missing combinatorial multiplier.

The BCR model follows the same high-level route but inserts the missing structural object: the feasible assignments of the middle $3 \times 3$ block. Once this multiplier (56) is made explicit, the rest of the derivation becomes mechanical. The important qualitative point is that BCR is not "shorter because it skips detail"; it is shorter because it allocates detail to the only place where it matters. In this example, concision and correctness align because BCR eliminates interpretive chatter while preserving the crucial combinatorial factor that the baseline overlooked.

**Example 9 (Table 16): Telescoping Log Product (65.3% reduction, correct → correct).** This example isolates pure compression without any accuracy change. Both models eventually discover the same decomposition: use the power rule for logarithms, separate the product into a rational term and a logarithmic term, telescope both pieces, and multiply the results. What differs is the amount of meta-reasoning expended before the derivation begins. Nearly all of the extra baseline tokens are spent on interpretation loops about notation and product structure rather than on the telescoping algebra itself.

The baseline repeatedly re-parses the statement, questions whether the expanded product is merely illustrative or structurally important, and spends a long prefix of the trace deciding how to interpret notation that does not actually change the solution strategy. BCR still contains some setup language, but it commits much earlier to the right abstraction. This suggests that on already-solved problems, BCR's main benefit is not discovering new mathematics; it is reducing "problem-understanding overhead" that contributes many tokens while adding no new information once the telescoping structure is recognized.

**Example 10 (Table 17): Nine-Point Circle Arc Geometry (23.2% reduction, no answer → correct).** The third AIME example is useful because it shows that BCR can help even when the token reduction is moderate rather than dramatic. The problem asks for a weighted combination of arc lengths on the circumcircle of the medial triangle. The baseline correctly recognizes the nine-point-circle setting and recalls several relevant angle-to-arc relationships, but it never stabilizes these local facts into a single consistent set of target arcs. The trace ends in unresolved discussion about whether arcs such as 5.0pt$24.88pt\overset{\frown}{DG}$ and 5.0pt$\overset{\frown}{DE}$ should coincide, and no valid boxed answer is produced. The token-heavy failure mode here is theorem accumulation without binding: the model keeps generating related geometric facts, but never commits them to the exact variables in the question.

The BCR trace is not perfectly elegant here; it still mixes theorem recall with approximate geometric estimation. But unlike the baseline, it does something qualitatively crucial: it commits. After identifying 5.0pt$\overset{\frown}{DE} = 72°$, it settles on 5.0pt$\overset{\frown}{HJ} = 24°$ and 5.0pt$\overset{\frown}{FG} = 72°$, then cleanly evaluates the requested linear combination to obtain 336. This example therefore illustrates a different Qwen-side compression mechanism from the previous two: BCR is not merely shortening successful derivations, but helping the model exit a long geometric deliberation with a coherent final synthesis.

*Table 15.* **Qwen AIME 2025, Example 8: Sudoku-Style Counting** (43.2% token reduction). The baseline omits a key combinatorial factor; BCR restores it and obtains the correct answer.

| **Qwen3-4B-Thinking-2507 (31,362 tokens)** | **BCR-Qwen3-4B (17,805 tokens)** |
|---|---|
| **Problem:** Count the valid $3 \times 9$ Sudoku-style fillings and write the total as $p^a q^b r^c s^d$; compute $pa + qb + rc + sd$. 
 **Ground Truth:** 81 | |
| *Uses $N = 9! \cdot 2^3 \cdot (3!)^3$ and factors this incomplete count.* *Most extra tokens are spent re-describing the grid geometry and block layout before the counting argument actually starts.* 
 *The actual error is structural: the trace never explicitly counts the 56 feasible middle-block assignments, so a long derivation is built on the wrong combinatorial factor.* 
 *Concludes $N = 2^{13} \cdot 3^7 \cdot 5 \cdot 7$, hence $2 \cdot 13 + 3 \cdot 7 + 5 + 7 = 59$.* | **Adds the missing structural multiplier: 56 feasible middle-block assignments.** 
 $N = 9! \cdot 56 \cdot (3!)^3 \cdot (3!)^3$. 
 **Thus $N = 2^{16} \cdot 3^{10} \cdot 5 \cdot 7^2$, so $2 \cdot 16 + 3 \cdot 10 + 5 + 14 = 81$.** |
| **Answer:** 59 ✗ | **Answer:** 81 ✓ |

*Table 16.* **Qwen AIME 2025, Example 9: Telescoping Log Product** (65.3% token reduction). Both models are correct; BCR removes prolonged setup chatter and preserves only the core derivation.

| **Qwen3-4B-Thinking-2507 (22,817 tokens)** | **BCR-Qwen3-4B (7,921 tokens)** |
|---|---|
| **Problem:** Evaluate $\prod_{k=4}^{63} \frac{\log_k(5^{k^2-1})}{\log_{k+1}(5^{k^2-4})} = \frac{m}{n}$ and find $m + n$. 
 **Ground Truth:** 106 | |
| *Long re-parsing preamble with repeated "wait/no/hold on" cycles before simplification.* 
 *The token overhead comes mainly from notation interpretation loops, not from the telescoping algebra itself.* 
 Eventually derives $\prod \frac{k^2-1}{k^2-4} = \frac{31}{13}$ and $\prod \frac{\log_k 5}{\log_{k+1} 5} = 3$. | Directly applies $\log_b(a^c) = c \log_b(a)$ and splits the product into: 
 **(i) rational telescoping $\rightarrow \frac{31}{13}$, (ii) logarithmic telescoping $\rightarrow 3$.** 
 Combines to $\frac{93}{13}$ and returns 106 . |
| **Answer:** 106 ✓ | **Answer:** 106 ✓ |

*Table 17.* **Qwen AIME 2025, Example 10: Nine-Point Circle Arc Geometry** (23.2% token reduction). The baseline recalls relevant geometric facts but never consolidates them into a final answer; BCR commits to a coherent arc decomposition and finishes.

| **Qwen3-4B-Thinking-2507 (32,215 tokens)** | **BCR-Qwen3-4B (24,754 tokens)** |
|---|---|
| **Problem:** In $\triangle ABC$ with angles $84°, 60°, 36°$, let $D, E, F$ be side midpoints and $G, H, J$ be additional intersections on the circumcircle of $\triangle DEF$. Compute $5.0 \text{pt} \overset{\frown}{DE} + 25.0 \text{pt} \overset{\frown}{HJ} + 35.0 \text{pt} \overset{\frown}{FG}$. 
 **Ground Truth:** 336 | |
| Recognizes the nine-point-circle setting and proposes identities such as 
 *$5.0 \text{pt} \overset{\frown}{DG} = 72°$, $5.0 \text{pt} \overset{\frown}{EH} = 120°$, $5.0 \text{pt} \overset{\frown}{FJ} = 168°$* 
 *Most extra tokens come from theorem-recall and arc-matching loops: the model keeps generating related geometric facts without consolidating them into the three arcs actually required.* 
 *It gets stuck on whether these are the same arcs the problem asks for, and no boxed final answer appears.* | Settles on **$5.0 \text{pt} \overset{\frown}{DE} = 72°$, $5.0 \text{pt} \overset{\frown}{HJ} = 24°$, $5.0 \text{pt} \overset{\frown}{FG} = 72°$** and then computes 
 $72 + 2 \cdot 24 + 3 \cdot 72 = 336$. |
| **Answer:** None ✗ | **Answer:** 336 ✓ |

## C.3.2. AMC 2023 Examples

We next present three AMC 2023 examples spanning recurrence extraction, sign analysis, and complex numbers. Because AMC problems are typically shorter than AIME or BRUMO, the qualitative contrast here is less about discovering entirely different solution paths and more about whether the model drifts into a "teaching mode" that is useful for a human audience but unnecessary for reward optimization. The Qwen-based examples show that BCR consistently suppresses this tendency. On the hardest AMC case, it prevents full degeneration; on the other two, it compresses long explanatory traces into short, invariant-driven arguments.

**Example 11 (Table 18): Triangular Array Sum (69.0% reduction, no answer → correct).** This is the AMC analogue of the long-failure cases in C.2. The baseline begins sensibly by tabulating the first few row sums and looking for a pattern, but it never transitions to a stable closed-form derivation. Instead, the trace eventually degenerates into long repeated digit strings and ends without a valid boxed answer. The token increase is driven by repeated small-case recomputation after the recurrence structure is already recoverable.

BCR keeps the same initial instinct but compresses it into a mathematically sufficient derivation. Once the row-sum recurrence $S(k) = 2S(k-1) + (k-2)$ is identified, the model moves immediately to the closed form $S(k) = 2^k - k$ and reduces the task to a one-line modular arithmetic computation. The important point is that the savings do not come from omitting the recurrence; they come from refusing to continue generating after the recurrence has already solved the problem.

**Example 12 (Table 19): Sign of a High-Multiplicity Polynomial (54.1% reduction, correct → correct).** In this problem, both models are correct and use the same underlying fact: crossing a root of odd multiplicity flips sign, while crossing an even-multiplicity root preserves sign. The baseline, however, expresses this insight in a verbose interval-by-interval narration that repeatedly restates the same invariant. This is a textbook token-overhead case: once parity is known, the remaining narration is almost entirely redundant bookkeeping.

BCR compresses the argument into exactly the right state representation: a parity table over the roots. Once the initial sign on $(-\infty, 1)$ is fixed, the rest of the solution is a deterministic scan over odd and even multiplicities. This is a good example of what "syntactic compression" means for a stronger model: the mathematics is unchanged, but the surface realization becomes much denser because repeated verbal bookkeeping is replaced by a compact structural summary.

**Example 13 (Table 20): Complex Equation with Conjugation (56.2% reduction, correct → correct).** The final AMC example shows a cleaner version of pedagogical narration removal. Both models solve $z^5 = \overline{z}$ by moving to polar form, isolating the modulus cases $r = 0$ and $r = 1$, and then recognizing that the unit-modulus branch reduces to the sixth roots of unity. The baseline is correct, but it spends many tokens restating familiar facts about conjugates, arguments, and distinctness of the resulting roots. Here the extra tokens come almost entirely from tutorial-style explanation rather than from mathematical search.

The BCR model reaches the same conclusion with much tighter exposition. After deriving $\overline{z} = 1/z$ on the unit circle, it immediately reduces the problem to $z^6 = 1$ and counts the six roots together with $z = 0$. The value of this example is that it shows BCR's effect even when the task is already easy for the base model: the learned compression policy suppresses explanatory surplus without weakening the actual mathematical chain of reasoning.

*Table 18.* **Qwen AMC 2023, Example 11: Triangular Array Sum** (69.0% token reduction). The baseline degenerates without a valid answer; BCR solves the recurrence and finishes normally.

| **Qwen3-4B-Thinking-2507 (32,577 tokens)** | **BCR-Qwen3-4B (10,096 tokens)** |
|---|---|
| **Problem:** In a modified Pascal-like triangular array, find the units digit of the sum of the 2023 numbers in row 2023. **Ground Truth:** 5 | |
| Starts correctly with row-sum exploration, then *the trace keeps recomputing small rows instead of collapsing early to a recurrence, so token usage grows before any real progress is made.* *output collapses into extremely long repeated digit sequences* *and no valid final boxed answer is produced.* | Derives $S(k) = 2S(k-1) + (k-2)$ and solves it as $S(k) = 2^k - k$. Then computes $2^{2023} \equiv 8 \pmod{10}$ and $(8-3) \bmod 10 = 5$. |
| **Answer:** None ✗ | **Answer:** $\boxed{5}$ ✓ |

*Table 19.* **Qwen AMC 2023, Example 12: Polynomial Sign Counting** (54.1% token reduction). BCR condenses interval-by-interval narration into a short parity-based sign table.

| Qwen3-4B-Thinking-2507 (21,987 tokens) | BCR-Qwen3-4B (10,082 tokens) |
|---|---|
| **Problem:** For $P(x) = \prod_{k=1}^{10}(x-k)^k$, on how many of the 11 open intervals between roots is $P(x)$ positive? **Ground Truth:** 6 | |
| *Narrates each interval separately, repeatedly restating which root crossings flip the sign and which do not.* *The token-heavy part is not the mathematics; it is the repeated interval-by-interval bookkeeping after the odd/even multiplicity invariant is already known.* | Builds the sign pattern directly from multiplicity parity: **odd multiplicity roots flip sign; even multiplicity roots preserve sign.** Positive intervals are $(1,2), (2,3), (5,6), (6,7), (9,10), (10,\infty)$, so the count is 6. |
| **Answer:** $\boxed{6}$ ✓ | **Answer:** $\boxed{6}$ ✓ |

*Table 20.* **Qwen AMC 2023, Example 13: Complex Equation with Conjugation** (56.2% token reduction). Both models are correct; BCR strips away pedagogical narration and keeps only the polar-form argument.

| Qwen3-4B-Thinking-2507 (11,599 tokens) | BCR-Qwen3-4B (5,082 tokens) |
|---|---|
| **Problem:** How many complex numbers satisfy the equation $z^5 = \overline{z}$? **Ground Truth:** 7 | |
| *Uses polar form correctly but spends many tokens re-explaining conjugates, arguments, and why the sixth roots are distinct.* *This is mostly pedagogical overhead: once the problem is reduced to $r=0$ or $z^6 = 1$, the remaining narration adds cost without changing the solution.* | Moves quickly to the two modulus cases: $r = 0 \Rightarrow z = 0$, **and** $r = 1 \Rightarrow \overline{z} = 1/z$, **so** $z^6 = 1$. Thus there are $1 + 6 = 7$ solutions. |
| **Answer:** $\boxed{7}$ ✓ | **Answer:** $\boxed{7}$ ✓ |

## C.3.3. BRUMO 2025 EXAMPLES

The BRUMO 2025 examples are the most informative for the 4B model because they separate three qualitatively distinct gains. First, BCR avoids catastrophic overthinking on long expectation problems. Second, it finds cleaner structural decompositions on graph-style reasoning tasks. Third, even when both models are correct, it substantially shortens epistemic elimination chains by organizing the candidate set earlier. Together, these examples show that the Qwen-based "free lunch" is not confined to one problem type: it appears in probability, graph structure, and logic-heavy deduction.

**Example 14 (Table 21): Card-Draw Stopping Time (66.3% reduction, no answer → correct).** This is the strongest BRUMO failure-to-success transition in the Qwen section. The baseline starts from a natural but dangerous strategy: enumerate states or relative orderings of first Ace/King/Queen appearances and try to average over them explicitly. That approach is mathematically viable in principle, but in practice it expands into a very long case analysis that never converges to a finished expression. The token explosion is therefore combinatorial: explicit order enumeration grows much faster than the compact expectation identity needed for the final answer.

BCR replaces this with a much more stable abstraction. By reasoning in terms of first-hit positions for nested target sets, it can write the stopping time expectation directly as an inclusion-exclusion expression over expected minima. Once this representation is adopted, the computation becomes short and linear. The example therefore illustrates a recurring 4B pattern: BCR does not merely "stop early"; it steers the model toward representations whose solution length is intrinsically shorter.

**Example 15 (Table 22): Party Friend/Enemy Graph (45.9% reduction, incorrect → correct).** The baseline understands that the condition "the friend of an enemy is an enemy" imposes strong global structure, but it never fully crystallizes what that structure is. Instead, it reasons with degree bounds and tests candidate values of $n$, eventually concluding too early that only 13 and 24 are feasible. Most of the wasted tokens come from local case testing before the global clique decomposition is made explicit.

The BCR solution makes the hidden invariant explicit: the friendship graph must decompose into equal cliques. Once this is stated, the rest of the problem collapses to the divisor identity $n = 12 + \frac{12}{k-1}$. That single structural move converts a diffuse graph search into a finite arithmetic enumeration. This is one of the cleanest examples in the appendix where shorter reasoning is not only cheaper but also conceptually sharper.

**Example 16 (Table 23): Multi-Agent Deduction with Four Bears (59.6% reduction, correct → correct).** The final BRUMO case is included to show that BCR also compresses successful long-form elimination. Both models solve the puzzle by propagating the sequence of public statements and shrinking the feasible set of tuples $(A, B, C, D)$. The baseline is correct, but it revisits several candidate sets multiple times and only late in the trace organizes them around Druno's final uniqueness requirement. The dominant token cost is repeated regeneration of the same small candidate family rather than new logical progress.

BCR reaches the same answer by turning the remaining ambiguity into a compact table much earlier. After fixing $C = 2$, it enumerates the five viable $(A, B, D)$ combinations and observes that only one gives a unique value of $D$. This is exactly the kind of reasoning that benefits from BCR's training setup: once the candidate space has been compressed to a small table, any additional prose is pure overhead, so the learned policy is to stop.

*Table 21.* **Qwen BRUMO 2025, Example 14: Card-Draw Stopping Time** (66.3% token reduction). The baseline stalls in state enumeration; BCR switches to a first-hit formulation and reaches the exact expectation.

| Qwen3-4B-Thinking-2507 (32,645 tokens) | BCR-Qwen3-4B (11,007 tokens) |
|---|---|
| **Problem:** In a 54-card deck with two jokers, find the expected number of draws until at least one Ace, one King, and one Queen have appeared. **Ground Truth:** $\frac{737}{39}$ | |
| *Large state and permutation enumeration expands to 32K tokens and never yields a valid boxed answer.* *The token explosion is caused by explicit order/case enumeration over first-hit events, which grows much faster than the final inclusion-exclusion formula actually requires.* | Introduces first-hit positions for progressively larger target sets and applies $\mathbb{E}[T] = 3\mathbb{E}[M] - 3\mathbb{E}[L] + \mathbb{E}[R]$. With $\mathbb{E}[M] = \frac{55}{5}$, $\mathbb{E}[L] = \frac{55}{9}$, and $\mathbb{E}[R] = \frac{55}{13}$, $\mathbb{E}[T] = \frac{737}{39}$. |
| **Answer:** None × | **Answer:** $\boxed{\dfrac{737}{39}}$ ✓ |

Taken together, these nine Qwen-based examples show a section-wide pattern closely paralleling Appendix C.2: BCR-

*Table 22.* **Qwen BRUMO 2025, Example 15: Party Friend/Enemy Graph** (45.9% token reduction). BCR uses clique decomposition to enumerate all feasible values of $n$; the baseline undercounts the solution set.

| Qwen3-4B-Thinking-2507 (26,146 tokens) | BCR-Qwen3-4B (14,139 tokens) |
|---|---|
| **Problem:** Each guest has exactly 12 enemies and "the friend of an enemy is an enemy." Find the sum of all possible values of $n$. **Ground Truth:** 100 | |
| Builds graph-theoretic constraints but concludes only $n = 13$ *and* $n = 24$, so the final answer is 37. *Most extra tokens come from testing local degree bounds and candidate sizes one by one instead of identifying the global clique decomposition early.* | Represents the friend graph as a disjoint union of $k$ equal cliques of size $s$: $n = ks$ and $n - s = 12$, so $n = \frac{12k}{k-1} = 12 + \frac{12}{k-1}$. **Since $k - 1$ must divide 12, the feasible values are** $13, 14, 15, 16, 18, 24$**, whose sum is** $100$**.** |
| **Answer:** $\boxed{37}$ $\times$ | **Answer:** $\boxed{100}$ ✓ |

*Table 23.* **Qwen BRUMO 2025, Example 16: Four-Bear Deduction Puzzle** (59.6% token reduction). Both models are correct; BCR organizes the surviving candidate tuples into a compact uniqueness table much earlier.

| Qwen3-4B-Thinking-2507 (28,830 tokens) | BCR-Qwen3-4B (11,648 tokens) |
|---|---|
| **Problem:** Four bears hold positive integers summing to 17 and make sequential public statements; determine the product of their four numbers. **Ground Truth:** 160 | |
| *Eventually identifies the valid tuple* $(2, 5, 2, 8)$ *but spends many tokens repeatedly rebuilding the remaining candidate sets before testing Druno's uniqueness condition. The token-heavy step is repeated candidate-set reconstruction: the same small family of tuples is regenerated several times instead of being frozen into one compact table.* | After deducing $C = 2$, quickly tabulates the surviving possibilities: $(2, 5, 8), (2, 8, 5), (5, 5, 5), (5, 8, 2), (8, 5, 2)$ and observes that only $D = 8$ is unique, yielding $(A, B, C, D) = (2, 5, 2, 8)$ and product 160. |
| **Answer:** $\boxed{160}$ ✓ | **Answer:** $\boxed{160}$ ✓ |

Qwen is not simply more terse, but systematically better at converting long exploratory reasoning into compact structural arguments. On the easiest cases, this appears as narration removal; on mid-difficulty cases, as earlier commitment to the correct abstraction; and on the hardest cases, as outright prevention of unfinished or degenerate traces.

## C.4. Summary of Qualitative Patterns

We conclude the qualitative analysis with a systematic summary of the compression patterns observed across all seven examples, synthesizing the individual findings into broader insights about BCR's effect on reasoning behavior.

**Summary Table.** Table 24 aggregates the token counts, compression ratios, and dominant mechanisms for each example.

*Table 24.* Summary of qualitative examples across three benchmarks and four compression mechanisms. Reductions measured as percentage decrease in token usage relative to the baseline.

| Ex. | Source | Base Tok | BCR Tok | Reduction | Dominant Mechanism |
|---|---|---|---|---|---|
| 1 | AIME 2025 | 7,211 | 3,639 | 49.5% | Metacognitive loop elimination |
| 2 | AIME 2025 | 6,676 | 4,164 | 37.6% | Redundant verification removal |
| 3 | AIME 2025 | 3,530 | 2,216 | 37.2% | Strategy exploration reduction |
| 4 | AMC 2023 | 10,122 | 3,261 | 67.8% | Error prevention via conciseness |
| 5 | AMC 2023 | 3,464 | 1,425 | 58.9% | Pedagogical narration removal |
| 6 | AMC 2023 | 4,329 | 2,182 | 49.6% | Redundant verification removal |
| 7 | BRUMO 2025 | 32,677 | 2,692 | 91.8% | Catastrophic degeneration prevention |
| **Average** | | 9,716 | 2,797 | **56.1%** | — |

**Key Insights.** Several patterns emerge from the cross-example analysis:

*(1) Compression scales with baseline verbosity.* The token reduction percentage correlates strongly with the baseline's absolute token count. Examples where the baseline is most verbose (Ex. 7: 32,677 tokens; Ex. 4: 10,122 tokens) show the largest reductions (91.8% and 67.8% respectively), while examples where the baseline is already relatively concise (Ex. 3: 3,530 tokens) show more modest reductions (37.2%). This suggests that BCR is particularly effective at compressing the "tail" of verbose outputs—the extreme cases that dominate inference costs in practice.

*(2) All compression is syntactic, not semantic.* Across all seven examples, the mathematical content of BCR's solutions is identical to or richer than the baseline's. BCR never skips a necessary factorization, omits a boundary check, or shortcuts a logical deduction. The compression operates exclusively at the syntactic level: eliminating self-doubt phrases, removing pedagogical narration, avoiding strategy exploration dead ends, and preventing degenerate generation. This syntactic-only compression explains why BCR can achieve large token reductions without sacrificing accuracy.

*(3) Efficiency can improve accuracy (Ex. 4).* The complex analysis example demonstrates that verbosity is not merely wasteful—it can be actively harmful. When the baseline explores multiple approaches over 10,000 tokens, it creates more opportunities to confuse itself, ultimately arriving at an incorrect answer. BCR's concise, committed reasoning avoids this failure mode. While this is a single example, the quantitative results in the main paper (accuracy improvements on 3/5 benchmarks, with only modest decreases on the other two) confirm that this pattern generalizes beyond individual examples.

*(4) Different mathematical domains trigger different compression mechanisms.* Algebra problems primarily trigger metacognitive loop elimination (the model stops second-guessing its factorizations). Geometry problems primarily trigger redundant verification removal (the model stops re-checking coordinate computations). Number theory problems trigger strategy exploration reduction (the model identifies the correct approach faster). This diversity suggests that BCR learns a general principle of token economy rather than domain-specific compression heuristics.

*(5) The average reduction (56.1%) is consistent with quantitative results.* The mean token reduction across our seven examples (56.1%) falls squarely within the range reported in the main paper (39.8–62.6% across five benchmarks). This consistency between qualitative examples and quantitative results reinforces our confidence that the examples are representative of BCR's general behavior, not cherry-picked outliers.

**Broader Implications.** These qualitative findings support our central thesis: LLMs possess latent high-density reasoning modes that are normally suppressed by single-problem training. Standard RL training with accuracy-only rewards inadvertently incentivizes verbose deliberation—the model learns that "showing work" correlates with higher reward, even when much of that work is redundant. BCR's multi-problem training breaks this spurious correlation by creating a setting where redundant tokens impose a real cost (reduced budget for subsequent problems), naturally selecting for information-dense reasoning strategies.

