# OpenReview forum: "Batched Contextual Reinforcement"
_ICML.cc/2026/Conference — ICML 2026 regular_

### Official Review · Reviewer_e4gi · 2026-03-09

**Soundness:** 2
**Presentation:** 2
**Significance:** 3
**Originality:** 3
**Overall Recommendation:** 4
**Confidence:** 4

**Summary:**

- Chain-of-thought reasoning can significantly improve performance but often incurs high computational costs due to long reasoning traces.
- The paper proposes a method to train models to produce token-efficient reasoning while maintaining strong task performance.
- The key idea is to incentivize the model to solve multiple tasks within a single context window, creating competition for a shared token budget and encouraging concise reasoning.

**Compliance With Llm Reviewing Policy:**

Affirmed.

**Final Justification:**

This paper proposes a simple and elegant method for improving token efficiency in reasoning models by training on multiple problems within a shared context window. The core idea of inducing an implicit token budget through batched prompts is novel and conceptually appealing, and the empirical results demonstrate substantial reductions in token usage while maintaining or even improving accuracy on several benchmarks. The motivation is clear, and the method is easy to understand and implement.

From a soundness perspective, I initially had concerns regarding the potential train–test distribution shift (multi-problem training vs. single-problem evaluation), the limited scope of training (single dataset and model), and the lack of clarity around baseline comparability. The authors’ rebuttal partially addresses these issues. In particular, the addition of experiments on a stronger base model (Qwen3-4B) strengthens the claim that the method is not tied to a specific model, and the reward ablation helps clarify aspects of the training objective. The commitment to improving transparency around training details and prompts is also appreciated.

However, some concerns remain only partially resolved. The train–test discrepancy is better motivated but not fully analyzed empirically, and baseline comparability is still somewhat unclear without detailed aligned training settings. Additionally, generalization across training datasets is not empirically demonstrated.

Despite these limitations, I find that the strengths of the paper outweigh the weaknesses. The idea is original, practically relevant, and supported by reasonably strong empirical evidence, including additional results provided in the rebuttal. The method has the potential to be impactful for improving the efficiency of reasoning models, which is an important problem for the community.

**Key Questions For Authors:**

- Is there a more rigorous analysis of the training–evaluation distribution discrepancy? For example, how sensitive is performance to different orderings of the prompts within a batch?
- Are there experiments using base models other than JustRL-DeepSeek-1.5B?
- Are there experiments using training datasets other than DeepMath-103K?
- Are there ablations on the reward shaping weights used in the reward function?
- What are the exact training settings for the baseline methods used in the comparisons? For example:
  - How many training steps were used?
  - Are the baselines trained on top of the same base model?
  - Do they use the same response length cutoff or penalty during training?
  - What reinforcement learning algorithm do they use?
  - What training dataset are they trained on?

**Limitations:**

Yes

**Strengths And Weaknesses:**

## Soundness
- During training, the model is prompted with multiple problems within a shared context window, whereas evaluation is primarily conducted on single-problem prompts. This introduces a potential train–test distribution shift. It would strengthen the paper if the authors investigated the impact of this discrepancy more rigorously.
- The method is trained using only one dataset (DeepMath-103K) and one base model (JustRL-DeepSeek-1.5B), which limits the strength of claims about generalizability.
- It is not clear whether the baseline methods were trained under comparable conditions to BCR. Without clear information about training setups, there is a risk that differences in training procedures could introduce confounding factors.

## Presentation
- Minor formatting issue in the abstract: there is a missing space in “path.Empirically”.
- The motivation for the approach is clearly presented and easy to follow.
- Including the prompt template in the appendix is helpful. It would also be useful to include the full system prompt used during training.

## Significance
- The paper addresses an important problem for the community: reducing the inference cost of reasoning models while maintaining performance.

## Originality
- The idea of encouraging efficiency through a shared token budget across multiple tasks appears novel and creative.

---

> ### Author Rebuttal · Authors · 2026-03-31
>
> We thank the reviewer for recognizing the importance of token-efficient reasoning, the clarity of our motivation, and the novelty of encouraging efficiency through a shared token budget across multiple tasks. We address below the questions on train-test distribution shift, baseline comparability, and the scope of generalization across models.
>
> **1. training–evaluation length difference (S1-Q1)**
>
> Thank you for raising this point. The gain at $N=1$ is not a test-time formatting artifact from batched prompting. Rather, BCR induces a persistent shift toward higher-density reasoning. Empirically, even the base model shortens its outputs under multi-problem inference, suggesting that this efficient mode already exists latently before training. BCR then reinforces and stabilizes this behavior, allowing the learned compression to transfer back to ordinary $1\times$ inference. We will clarify this train-test connection in the revision.
>
> **2. Problem order sensitivity (S1-Q1)**
>
> During training, we deliberately shuffle problems within each batch, so the model does not rely on fixed order or semantically linked neighboring questions. This makes the batched context less order-dependent and keeps adjacent questions relatively irrelevant.
>
> This also connects to the optimization analysis in **Reviewer NyE8**. For the averaged BCR gradient direction $\overline{g}_N$, for any unit vector $u$,
> $$
> Var(u^\top \overline{g}_N)=\frac{\sigma_u^2}{N}\bigl(1+(N-1)\rho_u\bigr),
> $$
> where $\sigma_u^2$ is the single-problem gradient variance and $\rho_u$ is the cross-problem gradient correlation. Lower inter-problem correlation strengthens the variance-reduction effect of batching. From this perspective, shuffling is not only an implementation detail; it also helps avoid overly correlated batches that could weaken the stabilization effect. We will clarify this point in the revision and continue examining order-related effects experimentally.
>
> **3. Other base models (S2-Q2)**
>
> Thank you for this suggestion. To test generality beyond JustRL-DeepSeek-1.5B, we added experiments on **Qwen3-4B-Thinking-2507**. We observe the same overall pattern on this substantially stronger backbone: BCR consistently improves the accuracy-efficiency trade-off, suggesting that the method is not tied to a single base-model family.
>
> |  | AIME25 | AMC23 | Minerva | MATH-500 | Olympiad |
> | :--- | :--- | :--- | :--- | :--- | :--- |
> | Baseline | 70.0/20773 | 97.5/10457 | 66.2/4576 | 97.0/5136 | 83.3/13068 |
> | Ours | 83.3/17498 | 100.0/7128 | 68.4/3338 | 97.4/3713 | 85.2/10717 |
>
> **4. Other training datasets (S2-Q3)**
>
> Thank you for raising this question. We use **DeepMath-103K** because it provides a broad range of problem types and reasoning lengths, making it a practical starting point for grouped training and controlled analysis. This choice is not meant to suggest that BCR depends on a specific corpus. We will clarify this motivation in the revision and continue testing additional training corpora in follow-up experiments.
>
> **5. Reward-shaping weights (Q4)**
>
> Thank you for suggesting this ablation. We compared the default reward weights **2-1 (ours)** and **5-1**. Empirically, **5-1** yields a worse accuracy-efficiency trade-off, indicating that the format reward plays a functional role: it helps stabilize structured multi-problem generation, supports reliable answer extraction, and preserves accuracy under shared budget pressure.
>
> |  | AIME25 | AMC23 | Minerva | MATH-500 | Olympiad |
> | :--- | :--- | :--- | :--- | :--- | :--- |
> | 5-1 | 23.3/2905 | 80.0/2925 | 44.9/4939 | 88.0/2438 | 60.0/3207 |
> | 2-1 (Ours) | 33.3/3173 | 87.5/2637 | 48.5/2494 | 87.6/1868 | 62.9/2969 |
>
> **6. Exact baseline settings (S3-Q5)**
>
> Thank you for this suggestion. We will improve transparency by providing a linked summary table with the full settings of all compared baselines: https://anonymous.4open.science/r/Rebuttal_BCR-B7D0/BCR.pdf. We also note that the comparison is designed to be fair rather than selective: it includes general and math-specialized models, unconstrained and length-controlled methods, as well as adaptive-reasoning and RL-based approaches.
>
> **7. Reproducibility and formatting (P1, P3)**
>
> Thank you for pointing this out. We will improve reproducibility by including the full training system prompt and additional implementation details in the supplementary material: https://anonymous.4open.science/r/Rebuttal_BCR-B7D0/BCR.pdf. We will also fix the formatting issue in “path.Empirically” in the revision.
>
> Overall, we hope these clarifications and additional results better address the concerns on distribution shift, comparability, and generalization, while further strengthening the empirical support and reproducibility of BCR.

---

> > ### Author Rebuttal · Reviewer_e4gi · 2026-04-04
> >
> > I thank the authors for the detailed and thoughtful rebuttal. The additional experiments and clarifications have strengthened the paper in several important aspects.
> >
> > In particular, I appreciate:
> > - The inclusion of results on an additional base model (Qwen3-4B), which provides useful evidence that the proposed approach is not limited to a single model family.
> > - The added ablation on reward weights, which helps clarify the role of the reward design.
> > - The commitment to improving reproducibility by releasing more detailed training settings and the full system prompt.
> >
> > These additions increase my confidence in the empirical validity and practical relevance of the proposed method.
> >
> > That said, some of my original concerns are only partially addressed:
> >
> > - The potential train–test distribution shift (multi-problem training vs. single-problem evaluation) is discussed and better motivated, but would benefit from more direct empirical analysis to isolate its effect.
> > - The question of baseline comparability remains somewhat unclear. While the authors mention that they will provide a summary table of baseline settings, the rebuttal does not yet fully clarify whether all methods are compared under aligned training conditions.
> > - The generalization across training datasets is not empirically addressed, and remains an open question.
> >
> > Overall, I find the proposed idea to be novel, simple, and potentially impactful for improving reasoning efficiency. The additional evidence provided in the rebuttal strengthens the paper, and I am inclined to support acceptance. However, I encourage the authors to further clarify the above points in the final version to improve rigor and transparency.

---

> > > ### Author Response · Authors · 2026-04-06
> > >
> > > We are very grateful to the reviewer for the encouraging assessment and for raising the score. We deeply appreciate the recognition of the novelty, simplicity, and potential impact of BCR. In this response, we address the three remaining open points as carefully as possible.
> > >
> > > **1. Train–Test Distribution Shift.**
> > >
> > > We fully understand this concern and want to provide the most direct empirical evidence available. The key observation is that BCR's efficiency gains appear robustly at **N=1** (single-problem evaluation), which is the standard test setting — BCR achieves 40–62% token reduction on individual problems despite being trained at N=3. This transfer is not a formatting artifact: the base model also shortens outputs under batched inference, confirming that a latent compression mode pre-exists; BCR crystallizes it into a stable, transferable policy.
> > >
> > > To further substantiate this, we provide a Pareto frontier visualization tracking the 1× evaluation result across every training checkpoint on the Minerva benchmark (anonymous link: https://anonymous.4open.science/r/BCR-ICML-87C6/Minerva_Pareto.jpg). The trajectory shows monotonic improvement of the Pareto frontier throughout training — BCR consistently moves toward higher accuracy and lower token usage in the 1× regime simultaneously. This directly demonstrates that the train–test gap does not introduce a systematic bias; rather, the policy improvement learned under batched training transfers cleanly to single-problem inference at every stage of training. We will include this figure and the associated discussion in the revision.
> > >
> > > Additionally, Table 3 in the paper reports results across N ∈ {1,2,3,4,5} at test time. BCR outperforms the baseline at every group size, and the efficiency advantage grows as N increases — exactly the regime most relevant to practical deployment. This further supports the generality of the learned policy.
> > >
> > > **2. Baseline Comparability.**
> > >
> > > We appreciate the reviewer's continued attention to this point and want to be transparent. The baselines in our comparison span several distinct categories — general LLMs (Qwen3-1.7B), math-specialized models (STILL-3-1.5B, BroRL-1.5B, e3-1.7B), and adaptive reasoning models (ARM-3B, Thinker-Q1.5B) — each trained with different datasets, algorithms, and training durations as reported in their respective papers. We do not claim to control all of these variables across third-party models, and we believe this is standard practice in the field.
> > >
> > > For our own experiments, all BCR variants and ablations are implemented within the same GRPO framework, with hyperparameters held fixed except for the specific variable under investigation — this ensures that all internal comparisons are clean and controlled. We will incorporate a clearer summary table of baseline settings in the revision appendix to make this transparent to all readers.
> > >
> > > **3. Generalization Across Training Datasets.**
> > >
> > > Regarding generalization across training datasets: while we have proven that the BCR mechanism generalizes across different model scales (1.5B and 4B models), we acknowledge that improving a reasoning model inherently requires training data that matches the model's specific capability level. If a dataset is too simple or too misaligned, the model will not deepen its reasoning capacity.
> > >
> > > We fully agree that exploring how BCR interacts with diverse datasets is a vital next step. We will explicitly state this in Limitations section, where we will discuss the nuances of dataset selection and outline future experiments on scaling to larger models.
> > >
> > > Thank you again for pushing us to ensure the utmost rigor in this work; your meticulous review has directly improved the final quality of this paper.

---

### Official Review · Reviewer_MLqa · 2026-03-11

**Soundness:** 3
**Presentation:** 3
**Significance:** 3
**Originality:** 3
**Overall Recommendation:** 5
**Confidence:** 3

**Summary:**

This paper propose Batched Contextual Reinforcement (BCR), a training framework that applies reinforcement learning to llms by grouping multiple contextual tasks/questions into a single batch and optimizing rewards jointly. The method packages N questions together with system instructions and token constraints, computes an accuracy-based reward over extracted answers, adds a format reward, and optimizes the model using a GRPO training procedure. The proposed methods achieved on par performance on mathematical reasoning benchmarks with significant reduce in token usage.

**Compliance With Llm Reviewing Policy:**

Affirmed.

**Final Justification:**

I appreciate the follow-up answers. I have raised my score.

**Key Questions For Authors:**

See weakness

**Limitations:**

The paper doesn't include a limitation section.

**Strengths And Weaknesses:**

strength

The idea is simple and elegant and effective.

The paper is overall well-written and easy to follow.

weakness

The literature review is not comprehensive enough. In particular:

Closely related work on contextual bandits is absent.
Recent work on minimal-data or extreme-efficiency RL for reasoning [Reinforcement Learning for Reasoning in Large Language Models with One Training Example (2025)]

The token budget seems to be fixed across all main experiments. Since the token budget is described as the critical design choice enabling implicit length control, ablation studies varying the budget size are necessary.

The structured batch is formed as a partition of the training set. It would be interesting to include experiments with alternative batch constructions, such as:

Randomly sampled questions,
Repeated questions within a batch,
Mixtures of relevant and irrelevant questions,
Questions combined with irrelevant or distractor text.

---

> ### Author Rebuttal · Authors · 2026-03-31
>
> We thank the reviewer for highlighting that our idea is simple, elegant, and effective, and for noting that the paper is overall well-written and easy to follow. We respond below with additional clarification on related work, token-budget ablations, and alternative batch construction strategies.
>
> **1. Related work on contextual bandits and minimal-data RL (W1)**
> The current draft does not sufficiently position BCR within the broader literature on resource-constrained decision making. In the revision, we will expand the related-work section to make the connection to **contextual bandits with budget/knapsack constraints** explicit. We will also add relevant recent work on minimal-data RL for reasoning, including *Reinforcement Learning for Reasoning in Large Language Models with One Training Example (2025)*, which is relevant and currently missing.
>
> **2. Token-budget ablation (W2)**
> The token budget $B_{max}$ is indeed a central hyperparameter and should be examined directly. In our setup, $B_{max}$ is determined by estimating the average output length of the base model on a small sample of questions ($L_{avg}$), and then applying a compression target ratio $\lambda$ (default $\lambda \approx 0.5$), i.e.,
> $$
> B_{max} \approx N \times L_{avg} \times \lambda .
> $$
>
> To address this point, we conducted an additional budget sweep for the 1.5B model with $N=3$:
>
> | $B_{max}$ | AIME25 | AMC23 | Minerva | MATH-500 | Olympiad |
> | :--- | :--- | :--- | :--- | :--- | :--- |
> | **4096** | 26.7 / 2561 | 82.5 / 2135 | 46.0 / 2085 | 86.6 / 1524 | 61.1 / 2711 |
> | **5120 (Ours)** | 33.3 / 3173 | 87.5 / 2637 | 48.5 / 2494 | 87.6 / 1868 | 62.9 / 2969 |
> | **6144** | 30.0 / 3527 | 80.0 / 2701 | 44.9 / 2875 | 88.8 / 1874 | 63.4 / 3116 |
>
> These results show a clear trade-off: **4096** is too restrictive and hurts accuracy; **5120** gives the best balance between compression and performance; **6144** weakens the budget pressure and reduces the token-efficiency gain.
>
> We also evaluated the same idea on **Qwen3-4B-Thinking-2507** with **$B_{max}=8000$**:
>
> |  | AIME25 | AMC23 | Minerva | MATH-500 | Olympiad |
> | :--- | :--- | :--- | :--- | :--- | :--- |
> | Baseline | 70.0/20773 | 97.5/10457 | 66.2/4576 | 97.0/5136 | 83.3/13068 |
> | Ours | 83.3/17498 | 100.0/7128 | 68.4/3338 | 97.4/3713 | 85.2/10717 |
>
> We will include this token-budget ablation and discussion in the revision.
>
> **3. Alternative batch constructions (W3)**
> Alternative batch constructions are useful for clarifying which aspects of BCR are fundamental to the method and which may depend on the current implementation. In the current paper, we use stratified-difficulty grouping as a simple and stable default for the initial study, rather than as a claim that it is the only or best strategy. We will make this scope clearer in the revision.
>
> This question is also related to the optimization analysis discussed in **Reviewer NyE8**. For the averaged BCR gradient direction $\overline{g}_N$, we show that for any unit vector $u$,
> $$
> Var(u^\top \overline{g}_N)=\frac{\sigma_u^2}{N}\bigl(1+(N-1)\rho_u\bigr),
> $$
> where $\sigma_u^2$ is the single-problem gradient variance and $\rho_u$ is the cross-problem gradient correlation. This suggests that batch construction may affect training not only through the shared budget constraint, but also through the correlation structure among problems. When $\rho_u$ is smaller, the variance-reduction effect of batching is stronger; when problems are more redundant or closely correlated, this effect may be weaker.
>
> From this perspective, settings such as **randomly sampled questions**, **repeated questions within a batch**, **mixtures of relevant and irrelevant questions**, and **questions combined with irrelevant or distractor text** are worthwhile to study, as they may further clarify how inter-problem correlation relates to optimization behavior in BCR. We will clarify this connection in the revision, and we also plan to continue investigating these batch-construction settings experimentally in future work.
>
> Overall, the revision will:
> * Broaden the related-work discussion to include contextual bandits and recent minimal-data RL for reasoning;
> * Add an explicit token-budget sweep;
> * Clarify the role of batch construction and discuss additional settings.
>
> We hope these additions will clarify the scope, motivation, and empirical behavior of BCR, and will address the reviewer’s concerns more directly in the revised manuscript.

---

> > ### Author Rebuttal · Reviewer_MLqa · 2026-04-02
> >
> > Thank you for the authors' responses.  I am curious about the different batch construction and was expecting to see some preliminary results for this. And the submission doesn't include a limitation section. I maintain my score.

---

> > > ### Author Response · Authors · 2026-04-06
> > >
> > > We sincerely thank the reviewer for the continued engagement and for the highly insightful suggestion on batch construction variants. Following this suggestion, we conducted a systematic set of experiments that we believe meaningfully enrich the paper.
> > >
> > > **Batch Construction Experiments.**
> > >
> > > To address the reviewer's interest in alternative batch constructions, we implemented and evaluated four variants:
> > >
> > > - **Random_Sampled**: All questions are globally reshuffled and repacked into batches of size 3, completely reassigning within-batch composition while preserving total question count. All 3,000 resulting batches were verified to differ from the original groupings.
> > > - **Repeated_within_Batch**: Each batch consists of a single question duplicated across all 3 positions, isolating the effect of within-batch redundancy. All 3,000 batches were verified as fully repeated.
> > > - **Mixed_Relevant_Irrelevant**: Each batch contains 2 math questions from the original batch and 1 irrelevant question drawn from SQuAD (relevant_fraction = 0.5), randomly permuted. All 3,000 batches were verified to satisfy this composition.
> > > - **Distractor_Text**: Each original prompt is prepended with a short irrelevant passage from SQuAD, with the original question explicitly marked as the "Actual problem."
> > >
> > > Results (Accuracy / Tokens, 1.5B model):
> > >
> > > | Benchmark (Acc / Token) | Minerva | Olympiad | AMC23 | MATH-500 | AIME25 |
> > > | :--- | :--- | :--- | :--- | :--- | :--- |
> > > | **Baseline (Just RL)** | 43.4 / 5413 | 62.1 / 7017 | 85.0 / 5713 | 91.4 / 3099 | 40.0 / 8482 |
> > > | **BCR-JustRL-1.5B (Ours)** | 48.5 / 2494 | 62.9 / 2969 | 87.5 / 2637 | 87.6 / 1868 | 33.3 / 3173 |
> > > | **Mixed\_Relevant** | 31.6 / 3357 | 51.9 / 4496 | 80.0 / 4140 | 87.2 / 2222 | 33.3 / 6551 |
> > > | **Distractor\_Text** | 33.1 / 2850 | 53.3 / 3821 | 87.5 / 4171 | 86.2 / 1890 | 36.7 / 6644 |
> > > | **Repeated\_within\_Batch** | 30.1 / 2339 | 51.9 / 3731 | 87.5 / 3404 | 85.2 / 2029 | 30.0 / 3122 |
> > > | **Random\_Sampled** | 32.0 / 2334 | 53.0 / 2857 | 80.0 / 2143 | 86.2 / 1592 | 26.7 / 3010 |
> > >
> > > **Analysis.** These results collectively reveal an important pattern: all four alternative constructions effectively teach the model some degree of implicit compression (token counts drop across the board), confirming that the core BCR mechanism — shared budget pressure — is indeed the operative force. However, all four variants suffer notable accuracy degradation, especially visible on more challenging benchmarks (e.g., Minerva). We attribute this to **task interference**: each alternative introduces training signal that is qualitatively mismatched with high-difficulty mathematical reasoning. For instance, SQuAD-based irrelevant questions and repeated duplicates both dilute the reasoning depth required for competition-level problems, suppressing the model's capacity for deep logical inference. This is especially visible on Minerva, where accuracy drops substantially relative to our BCR setting, while easier benchmarks like AMC23 and MATH-500 are more robust to such interference.
> > >
> > > Notably, **Random_Sampled** shows the most aggressive token reduction but the largest accuracy drop — consistent with our optimization analysis: random regrouping creates highly heterogeneous within-batch difficulty, causing the model to adopt a conservative low-effort strategy rather than genuinely efficient reasoning.
> > >
> > > These findings reinforce that our original BCR design — stratified difficulty grouping with math-homogeneous batches — is not an arbitrary implementation choice but a principled decision. It achieves the best accuracy-efficiency Pareto point precisely because it maintains the integrity of the reasoning task while creating cross-problem competition. We will include this analysis, the four-variant table, and the task-interference discussion in the revision.
> > >
> > > **Limitation Section.** We will also add an explicit Limitations section in the revision, covering: (1) experiments are currently conducted on 1.5B and 4B models; scaling to larger models remains future work; (2) training is performed on DeepMath-103K; generalization across diverse training corpora is an open and interesting direction.
> > >
> > > We hope these additional experiments and the accompanying analysis address the reviewer's remaining questions. Given that the core suggestion has been implemented and yielded informative findings that strengthen the paper's positioning, we would be genuinely grateful if the reviewer would consider raising the rating.

---

### Official Review · Reviewer_NyE8 · 2026-03-12

**Soundness:** 2
**Presentation:** 2
**Significance:** 2
**Originality:** 2
**Overall Recommendation:** 4
**Confidence:** 3

**Summary:**

This paper studies the problem of overthinking in reasoning models trained with reinforcement learning. After RL post-training, many models produce very long chain-of-thought reasoning. This increases inference cost and sometimes even hurts performance. The paper proposes Batched Contextual Reinforcement (BCR). The main idea is simple: instead of training the model on one problem at a time, multiple problems are placed in the same prompt and solved within one shared output budget. Because the total token budget is limited, the model must allocate reasoning tokens across different problems. This creates an implicit competition between problems and encourages the model to remove redundant reasoning steps.

**Compliance With Llm Reviewing Policy:**

Affirmed.

**Final Justification:**

I agree that there are too much complicated algorithm design based on GRPO today. I really like the simple design behind BCR. I wil raise my rating.

**Key Questions For Authors:**

See above.

**Limitations:**

Yes.

**Strengths And Weaknesses:**

The proposed method is conceptually simple. It only modifies the prompt structure during training and does not require changes to the RL algorithm or model architecture. By letting multiple problems share a token budget, the model learns to reduce redundant reasoning. This provides a new perspective on controlling reasoning length.

However, the paper shows empirical improvements but does not deeply explain why the shared-budget setup leads to better reasoning efficiency. The method mainly changes the training setup and does not introduce a new theoretical framework or algorithmic improvement. During training, multiple problems are solved together, while inference is usually single-problem. The impact of this difference is not deeply analyzed.

---

> ### Author Rebuttal · Authors · 2026-03-31
>
> We thank the reviewer for recognizing the conceptual simplicity of our method and for highlighting that the shared-budget setup provides a new perspective on controlling reasoning efficiency. We further clarify the intuition behind this mechanism and discuss the train-test discrepancy between batched training and single-problem inference below.
>
> **1. Theoretical Framework: Why Shared Budgets Improve Efficiency**
>
> We agree that formalizing the underlying mechanics strengthens our algorithmic contribution. We have added a rigorous theoretical analysis that proves how BCR fundamentally alters the RL gradient dynamics compared to standard GRPO.
>
> * **Implicit Regularization via Budget Pressure:** We analyze the continuous relaxation of the token constraint. Let $l_k$ be the expected token length of the $k$-th problem, and $C_i = \sum_{j=1}^i l_j$ be the cumulative token usage up to problem $i$. By using a smooth surrogate $\\tilde{R} _N$ for the batch reward under the shared budget $B _{max}$ , we prove the expected reward gradient with respect to the token usage of *any* given problem is strictly non-positive:
>
>     We denote $P_k^{\\mathrm{bud}} = \\frac{\\partial\\mathbb{E}\[\\tilde{R}_N\]}{\\partial\\mathbb{E}\[l_k\]} $
>
>    Then,$P_k^{\\mathrm{bud}} \approx - \\frac{w_{acc}}{N} \\sum_{i \\ge k} \\mathbb{E} \left[c_i \\frac{1}{\\tau}\\sigma'\\left(\\frac{B_{max}-C_i}{\\tau}\\right)\right] \\le 0$
>
>   Here, $\sigma'$ is the positive derivative of the smooth threshold, and $c_i \ge 0$ relates to the correctness reward. This mathematically proves that verbose reasoning on *any* problem ($l_k$) directly penalizes the expected reward of all subsequent problems ($i \ge k$) within the shared context. This holistic structural pressure forces the model to optimize for overall information density without relying on explicit, unstable token penalties.
> * **Optimization Stability:** Furthermore, we prove BCR inherently reduces gradient variance. For any unit vector $u$, the variance of the BCR averaged gradient direction $\overline{g}_N$ scales as:
>     $$Var(u^\top \overline{g}_N) = \frac{\sigma_u^2}{N}(1 + (N-1)\rho_u)$$
>   Where $\sigma_u^2$ is the baseline variance and $\rho_u$ is the cross-problem gradient correlation. Assuming $\rho_u < 1$, the variance strictly decreases with larger $N$. This theoretically explains the highly stable optimization dynamics observed in our ablations (Figure 3), contrasting sharply with the collapse seen in explicit penalty methods.
>
> **2. Generalization and Practical Utility (Batch Training vs. 1x Inference)**
>
> The shift from multi-problem training to single-problem evaluation (1x inference) is indeed an important question. Rather than indicating a weakness, the successful transfer highlights a core strength of our method: the model does not simply overfit to the batched training format, but instead learns a persistent and concise reasoning policy that generalizes back to standard inference.
>
> Furthermore, 1x evaluation perfectly aligns with standard, real-world deployment scenarios. As detailed in our extensive multi-problem inference ablations (Table 3 and Table 6), the BCR-trained model consistently maintains a 40-60% token reduction across diverse inference group sizes (from 1x up to 5x). Retaining this extreme efficiency during 1x inference—where the external budget constraint is entirely absent—confirms that the efficiency gains reflect genuinely learned compression strategies that comfortably survive the train-test distribution shift.
>
> **3. Algorithmic Contribution**
>
> While BCR relies on the standard GRPO update rule, we respectfully argue that fundamentally redesigning the RL state-action formulation and reward environment to reliably induce emergent behaviors constitutes a significant algorithmic contribution. In a landscape dominated by brittle multi-stage curricula and complex auxiliary difficulty estimators, BCR provides a mathematically sound, single-stage framework that achieves state-of-the-art Pareto efficiency. We believe its architectural simplicity is its greatest strength, making it highly accessible for future reasoning research.
>
> We hope these theoretical additions and empirical clarifications address your concerns and highlight the depth of the BCR framework.

---

> > ### Author Rebuttal · Reviewer_NyE8 · 2026-04-02
> >
> > I agree that there are too much complicated algorithm design based on GRPO today. I really like the simple design behind BCR. I wil raise my rating.

---

> > > ### Author Response · Authors · 2026-04-02
> > >
> > > Thank you very much for the thoughtful engagement and for kindly supporting our work. We are grateful for the reviewer’s insightful feedback, which helped us make the paper more comprehensive and clearer in its motivation and contributions. We sincerely appreciate the reviewer’s time, careful reading and constructive support.

---

### Official Review · Reviewer_btij · 2026-03-13

**Soundness:** 3
**Presentation:** 3
**Significance:** 2
**Originality:** 3
**Overall Recommendation:** 4
**Confidence:** 3

**Summary:**

This paper introduces Batched Contextual Reinforcement (BCR), a training framework that aims to improve the token efficiency of language models in reasoning tasks without sacrificing accuracy. Instead of using explicit length penalties, BCR prompts the model to solve multiple problems within a single context under a shared token budget, creating an implicit pressure to be concise.

**Compliance With Llm Reviewing Policy:**

Affirmed.

**Final Justification:**

The rebuttal addressed your main concerns, so I will raise my rating.

**Key Questions For Authors:**

What is the influence of different Bmax on the model performance?
Can you provide more comparison between BCR and the other efficiency-oriented approach on JustRL-deepseek-1.5B?

**Limitations:**

No limitation section.

Suggestion:
See question.

**Strengths And Weaknesses:**

Strengths
1. Induces efficiency through a simple structural change without complex reward engineering or brittle length-penalty hyperparameters.
2. The design of Bmax allow the model to average the optimal reasoning length of several questions, which partly solve the question of assigning a fix target length for both easy and hard question in length reward.

Weaknesses
1. The paper dismisses all explicit length reward methods based on the failure of a single, simplistic implementation, which is not representative of more sophisticated and well-tuned length reward methods. This makes the method lack effective comparison.
2. While arguing for simplicity, the method's success still hinges on a hyperparameter (Bmax). Meanwhile the training group size N also have a large effect on the performance, which further increase the search space of the hyperparameter.
3. Lack of the curve between Bmax and model performance, which is critical to understand how the approach works.

---

> ### Author Rebuttal · Authors · 2026-03-31
>
> We thank the reviewer for recognizing the simplicity of BCR’s structural design and for noting that the shared budget offers a meaningful way to balance reasoning length across easy and hard questions. We address the concerns regarding comparisons with explicit length-control methods, the role of $B_{max}$, and the sensitivity to training group size below.
>
> **1. Explicit Length Penalties and SOTA Comparison (W1, Q2)**
>
> Our discussion of explicit length penalties in the manuscript serves to highlight the severe optimization instability inherent to static penalty configurations. As noted in our analysis, explicit penalties frequently induce catastrophic training collapse. While complex, multi-stage length-reward pipelines exist to mitigate this, they demand extensive and brittle engineering.
>
> To directly address your request for a comparison with well-tuned efficiency-oriented approaches, we direct your attention to the ProRL and BroRL models, which represent the state-of-the-art in complex length-control pipelines and are already included in our main results (Table 1).
>
> Crucially, both our 1.5B model and the BroRL-1.5B model originate from the exact same foundation: DeepSeek-R1-Distill-Qwen-1.5B. The ProRL framework employs an extensive 8-stage training strategy with a redesigned length-control mechanism. BroRL builds directly upon this heavily engineered pipeline. In contrast, we start with a model trained via simple single-problem GRPO and apply our batched training paradigm.
>
> Our single-stage method achieves highly competitive accuracy while demonstrating significantly superior token compression. This head-to-head comparison on the same base architecture unequivocally demonstrates that our simple structural incentive matches the efficacy of highly complex, multi-stage explicit penalty schedules while vastly reducing training complexity.
>
> **2. Influence and Selection of $B_{max}$ (W2, W3, Q1)**
>
> The selection of the token budget $B_{max}$ is a data-driven heuristic, not an arbitrary search. Because different base models possess varying inherent verbosity, we determine $B_{max}$ by sampling a small set of questions with the base model to find the average reasoning length ($L_{avg}$), and then applying a compression ratio $\lambda$ (e.g., $\lambda \approx 0.5$). Thus, $B_{max} \approx N \times L_{avg} \times \lambda$.
>
> To explicitly demonstrate the influence of $B_{max}$, we conducted additional ablations for our 1.5B model ($N=3$). **Due to space constraints, we have provided the full data table and our results on Qwen3-4B-Thinking in our response to Reviewer MLqa (W2).** Our analysis shows a clear optimization landscape: When $B_{max}$ is overly restrictive (4096), the model lacks sufficient capacity for intermediate reasoning steps, leading to a noticeable drop in accuracy on harder benchmarks. Conversely, when $B_{max}$ is too large (6144), the competitive pressure between problems diminishes, resulting in less effective token compression. The optimal value (5120) strikes the necessary balance, forcing the elimination of syntactic redundancy while fully preserving semantic reasoning steps.
>
> **3. Theoretical Foundation of the Implicit Budget (W1)**
>
> To mathematically ground why our method achieves stable compression without explicit penalties, we have formalized the gradient dynamics (to be added to Appendix). As shown in our newly derived Proposition 3, under a shared budget $B_{max}$, the expected reward gradient with respect to early token usage $l_k$ is strictly non-positive:
>
> We denote
> $$
> P_k^{\\mathrm{bud}} = \\frac{\\partial\\mathbb{E}\[\\tilde{R}_N\]}{\\partial\\mathbb{E}\[l_k\]}
> $$
>
> Then,
>
> $$P_k^{\\mathrm{bud}} - \\frac{w_{acc}}{N} \\sum_{i \\ge k} \\mathbb{E} \left[c_i \\frac{1}{\\tau}\\sigma'\\left(\\frac{B_{max}-C_i}{\\tau}\\right)\right] \\le 0$$
>
> This proves that the shared budget $B_{max}$ exerts a stable, implicit regularization pressure. Larger early-token usage mathematically induces negative reward pressure for subsequent problems without the need for manual, explicit token-penalty terms that often cause adversarial gradient collapse.
>
> **4. The Search Space of N (W2)**
>
> We extensively ablated the training group size $N \in \\{3, 4, 5\\}$ in Section 4.4.2 and Table 4 of our manuscript. The performance remains remarkably robust across these configurations, with $N=3$ consistently providing the best accuracy-efficiency trade-off for models at this scale.
>
> **5. Limitations**
>
> We will add a dedicated Limitations section to the revised manuscript, explicitly noting that our method's effectiveness is naturally bounded by the base model's inherent reasoning capacity and that extreme compression targets ($\lambda < 0.3$) may induce hallucination in smaller parameter models.

---

> > ### Author Rebuttal · Reviewer_btij · 2026-04-04
> >
> > Thanks for the response.  I will raise my rating.

---

> > > ### Author Response · Authors · 2026-04-04
> > >
> > > Thank you for acknowledging our rebuttal and for adjusting your rating!
> > >
> > > Your constructive feedback was incredibly helpful. The concerns you pointed out guided us to provide a clearer theoretical foundation and more comprehensive baselines, which significantly improved the overall quality and clarity of our paper. We are committed to incorporating all the discussions from this rebuttal phase into the camera-ready version.
> > >
> > > We deeply appreciate your time and your contribution to our research.

---

### Decision · Program_Chairs · 2026-04-30

**Decision:**

Accept (regular)

**Comment:**

This paper proposes a simple idea to improve reasoning efficiency with just the simple GRPO algorithm. Specifically, the method concatenates  a batch of problems into a single prompt and the models needs to solve all problems in a single response. As all the problem solutions share a total token budget here, the models would learn to implicitly allocate budget for problems of different difficulty. The reviewers gave a borderline positive scores of 4,4,4,5, the main strengths are that the proposed approach is novel, clever, and simple, without sophisticated modification of the GRPO algorithm. It is neat compared with previous approaches that do various length penalty. There are certain concerns remaining though after the rebuttal: (1) the original submission conducts all experiments just on one 1.5B model for only mathematical reasoning. While the authors added limited results of an additional 4B model during rebuttal, I still think this level of empirical experiments are too thin to well demonstrate generalization. (2) As mentioned by the reviewers, there is a train-test distribution mismatch here because the problems are not concatenated during test time. While empirical results in this paper do not show this as a problem, it is likely because that the base models already have strong potentials on these mathematical reasoning tasks, and then the distribution mismatch does not matter much, it may not be the case for other reasoning tasks though. Overall, I think this paper is borderline accept and I would recommend weak accept.